# Metabolic derangement in polycystic kidney disease mouse models is ameliorated by mitochondrial-targeted antioxidants

Nastaran Daneshgar [1], Andrew W. Baguley[1], Peir-In Liang[1,2], Fei Wu[3], Yi Chu[1], Michael T. Kinter[4], Gloria A. Benavides[5], Michelle S. Johnson[5], Victor Darley-Usmar[5], Jianhua Zhang [5], Kung-Sik Chan[3] & Dao-Fu Dai [1✉]

Autosomal dominant polycystic kidney disease (ADPKD) is characterized by progressively enlarging cysts. Here we elucidate the interplay between oxidative stress, mitochondrial dysfunction, and metabolic derangement using two mouse models of PKD1 mutation, PKD1$^{RC/null}$ and PKD1$^{RC/RC}$. Mouse kidneys with PKD1 mutation have decreased mitochondrial complexes activity. Targeted proteomics analysis shows a significant decrease in proteins involved in the TCA cycle, fatty acid oxidation (FAO), respiratory complexes, and endogenous antioxidants. Overexpressing mitochondrial-targeted catalase (mCAT) using adeno-associated virus reduces mitochondrial ROS, oxidative damage, ameliorates the progression of PKD and partially restores expression of proteins involved in FAO and the TCA cycle. In human ADPKD cells, inducing mitochondrial ROS increased ERK1/2 phosphorylation and decreased AMPK phosphorylation, whereas the converse was observed with increased scavenging of ROS in the mitochondria. Treatment with the mitochondrial protective peptide, SS31, recapitulates the beneficial effects of mCAT, supporting its potential application as a novel therapeutic for ADPKD.

[1] Department of Pathology, Carver College of Medicine, University of Iowa, Iowa City, IA, USA. [2] Department of Pathology, Kaohsiung Medical University Hospital, Kaohsiung Medical University, Kaohsiung, Taiwan. [3] Department of Statistics and Actuarial Science, College of Liberal Arts and Sciences, University of Iowa, Iowa City, IA, USA. [4] Aging & Metabolism Research Program, Oklahoma Medical Research Foundation, Oklahoma City, OK, USA. [5] Department of Pathology, Mitochondrial Medicine Laboratory, University of Alabama, Birmingham, AL, USA. ✉email: dao-fu-dai@uiowa.edu

Autosomal dominant polycystic kidney disease (ADPKD) is characterized by progressive development of multiple renal cysts that eventually result in end-stage renal disease (ESRD) in over 50% of affected patients[1]. Mutations in the *PKD1* gene, encoding polycystin-1 (PC-1), account for ~85% of ADPKD cases, with most of the remaining cases due to mutations in *PKD2*, which encodes polycystin-2[2]. PC-1 modulates mitochondrial function through sensing and responding to cellular $O_2$ levels[3] and also regulates cellular metabolism through one of its cleavage product[4]. The polycystins belong to a family of transient receptor potential (TRP) channels that are thought to form a complex to regulate $Ca^{2+}$ influx in response to extracellular mechanical stimuli in the primary cilium of kidney epithelial cells[5]. In the context of ADPKD, decreased intracellular calcium $[Ca^{2+}]$ downregulates $Ca^{2+}$-dependent phosphodiesterases and stimulates adenylyl cyclase 6 (AC6)[6], leading to elevated cAMP and subsequent activation of protein kinase A-mediated signaling[7]. This causes increased fluid secretion, proliferation, and disrupted tubular formation during cystogenesis[8]. Although cAMP-PKA signaling has been a major research focus leading to the discovery of Tolvaptan, which inhibits cell proliferation and fluid secretion by inhibiting the production of cAMP, other signaling pathways have been implicated in the pathogenesis of ADPKD, including activation of ERK and mTOR, downregulation of AMPK, and increased oxidative stress[1]. In addition, emerging evidence demonstrates that mitochondrial dysfunction and metabolic alterations are hallmark characteristics of ADPKD, implicating the role of metabolic regulation in ADPKD. However, the interplay between mitochondrial oxidative stress, metabolic derangement, and other related signaling pathways is not well-understood in the context of PKD1 mutations.

Homozygous PKD1 null mice are embryonic lethal; therefore, several other PKD1 mouse models have been developed to study human ADPKD. Among these, p.R3277C in the *PKD1* gene is a functionally hypomorphic mutation (partial loss-of-function), which is similar to mutations found in some ADPKD patients[9,10]. Mice with a knock-in of the R3277C allele in the homozygous state (designated RC/RC in this study) exhibit slowly progressive disease, whereas this mutant allele in combination with a deleted or null allele (RC/null) results in rapidly progressive PKD[9,10].

Catalase is a ubiquitous peroxisomal enzyme that catalyzes the decomposition of hydrogen peroxide into water and oxygen. We have previously shown that targeted overexpression of catalase in the mitochondria (mCAT) has a much greater beneficial effect in extending murine lifespan than overexpressing catalase targeted to the physiological site in the peroxisome. For example, catalase targeted to mitochondria (mCAT) decreases age-related degeneration in the brain, cardiac, and skeletal muscles and attenuates hypertensive cardiomyopathy and heart failure[11,12]. Therefore, to investigate the roles of mitochondrial $H_2O_2$ in the progression of PKD in RC/RC and RC/null mice, we generated adeno-associated virus serotype 9 carrying the mCAT transgene. We demonstrate that mitochondrial-targeted catalase suppresses cyst progression and kidney pathology that is at least partly driven by mitochondrial ROS in these two mouse models of PKD. The potential mechanisms are mediated by an increase in AMPK and the fatty acid oxidation pathway and mitigation of ROS-mediated ERK1/2 (extracellular signal-regulated protein kinases 1 and 2) phosphorylation.

## Results

**Mitochondrial dysfunction in PKD1 mouse models.** PKD1$^{RC/null}$ mice developed severe cystic enlargement that rapidly progressed to kidney failure around 20 days-of-age, whereas PKD1 RC/RC manifested as a slowly progressive cystic disease. To determine whether mitochondrial dysfunction occurs in these models, we measured mitochondrial complexes activity in protein homogenates from frozen kidney samples using extracellular flux analysis[13]. Lysates from 18 to 21 days old RC/null kidneys exhibited a dramatic and significant suppression of mitochondrial respiratory complexes I, II, III, or IV activities (Fig. 1a, d), while those from RC/RC kidneys exhibited a modest decrease in respiratory complexes activities (Fig. 1h, not significant). Citrate synthase activity was decreased by more than 50% in lysates from RC/null and ~30% in lysates from RC/RC kidneys compared to controls (Fig. 1b, f), suggesting a decrease in mitochondrial mass, whereas lactate dehydrogenase (LDH) activity, a cytosolic enzyme, remained unchanged in both RC/null and RC/RC kidneys compared to WT (Fig. 1c, g). The mitochondrial complex activities for RC/null kidneys decreased to less than 30% of the normal levels observed in WT kidneys (Fig. 1d). After normalization to citrate synthase, all respiratory complexes I−IV were decreased by ~50% in RC/null mice, indicating that in addition to a loss of mitochondrial mass, there were mitochondrial electron transport chain deficits associated with the PKD1 deletion (Fig. 1e). In RC/RC kidneys, there were trends of decreased complex III and IV ($p = 0.09$ and $p = 0.07$ respectively, Fig. 1h), with complex III exhibiting a slight insignificant decrease and complex IV exhibiting no change after normalization to citrate synthase (Fig. 1i), suggesting that the decreased mitochondrial mass was the main mechanism of mitochondrial impairment. However, given the variability, the sample size of 6, and $\alpha = 0.05$, the powers were 24% for complex I and ~50% for complexes II−IV, which were underpowered to detect the observed difference in RC/RC vs. WT kidneys.

To investigate the role of mitochondrial oxidative stress in the progression of PKD1 and the potential benefit of protecting mitochondria, we generated adeno-associated virus serotype 9 overexpressing catalase targeted to mitochondria using ornithine transcarbamylase leader sequence (AAV9-mCAT)[14,15] and injected it into RC/RC and RC/null mice (Fig. 2a). A low dose of AAV9-mCAT ($2 \times 10^9$/g) injection led to overexpression of catalase within the mitochondria of RC/RC mouse kidneys, as determined by immunofluorescence showing increased co-localization of mitochondrial-targeted catalase with VDAC, a mitochondrial marker (Fig. 2b, brown−orange color, middle) in most tubular epithelial cells. The successful delivery of AAV9-mCAT is evident in many tubular epithelial cells, including those adjacent to cysts in both RC/RC and RC/null kidneys (Fig. 2b, brown−orange color, middle and right). In contrast, injection with AAV9 vector alone showed lower levels of catalase in RC/RC mouse tubular epithelial cells, and the catalase staining did not overlap with VDAC (Fig. 2b, left), as expected, since the natural site of catalase is predominantly within the peroxisome[15]. To determine whether AAV9-mCAT differentially targets specific tubular segments, we performed co-staining of catalase, VDAC, and specific markers for proximal (Lotus Tetragonolobus Lectin, LTL) or distal tubules (Dolichos biflorus agglutinin, DBA). Our results showed comparable co-staining of catalase with LTL and DBA, without obvious difference between proximal and distal tubule in terms of mCAT delivery (Supplementary Fig. 1a−e). In addition to co-localization of catalase and VDAC (Fig. 2b), we fractionated fresh kidney tissues into mitochondrial-enriched and cytoplasmic fractions to further confirm the effective delivery of catalase targeted to the mitochondria. Immunoblotting of catalase showed increased catalase within the mitochondrial-enriched fractions in the AAV9-mCAT treated kidneys, with the ratio of catalase in mitochondrial/cytoplasmic fractions increased by greater than four-fold compared with that in mice treated with AAV9 vector-only (Supplementary Fig. 1f).

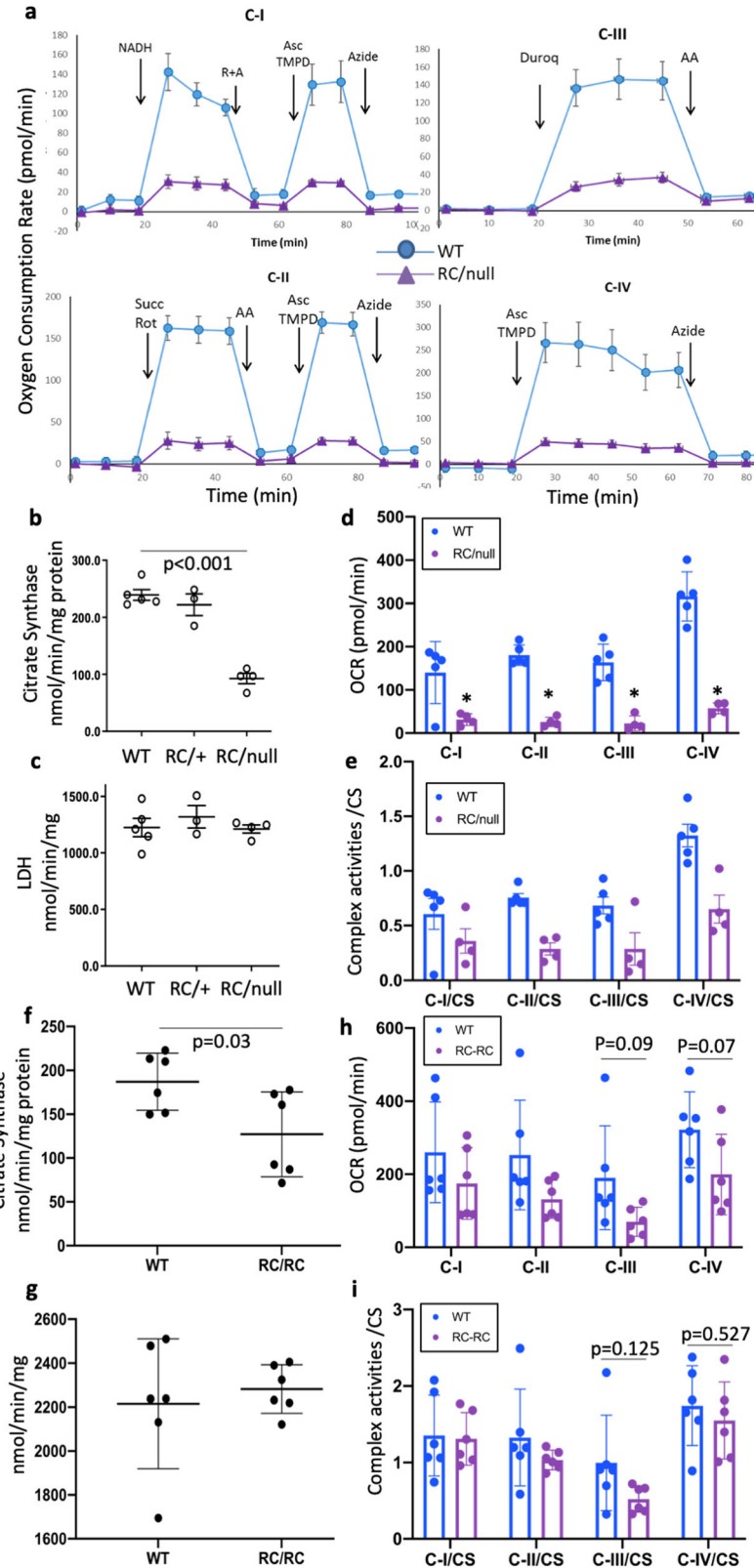

**Mitochondrial targeted catalase (mCAT) attenuates kidney pathologies in two mouse models of ADPKD.** RC/RC mice had enlarged kidneys at 6−7 month-of-age, and this was partially attenuated by low dose AAV9-mCAT injection (Fig. 3a). When normalized to body weight, kidney weight was significantly increased by ~60% in RC/RC mice compared with WT controls and this increase was attenuated by low dose AAV9-mCAT

(Fig. 3b). Heart weight normalized to body weight was also significantly increased in RC/RC mice, indicating cardiac hypertrophy, likely related to chronic hypertension, which is well documented in ADPKD patients[16]. This was completely prevented in RC/RC mice given a low dose injection of AAV9-mCAT (Fig. 3c). Blood urea nitrogen (BUN) was within normal range across all groups at 6−7 month-of-age (Fig. 3d).

**Fig. 1 Mitochondrial dysfunction in PKD1 mouse models. a** Representative traces of oxygen consumption rate (OCR) measured by a Seahorse XF analyzer, using various substrates to evaluate mitochondrial complex activity in frozen kidney lysates from WT and RC/null mice, including NADH (complex I), succinate (complex II), duroquinol (complex III) or Tetramethylphenylenediamine (complex IV, see method for details). Error bars represent the standard deviation of 3−5 repeated experiments. **b** Citrate synthase activity and **c** Lactate dehydrogenase (LDH) activity. **d** Oxygen consumption rates in kidney lysates from 5 WT, 4 RC/null, and 3 RC/+ mice as a measure of mitochondrial complex activity. **e** Mitochondrial complex activity normalized to citrate synthase activity in WT and RC/null kidney lysates. **f** Citrate synthase activity, **g** LDH activity, **h** oxygen consumption rates, **i** mitochondrial complex activity normalized to citrate synthase activity in frozen kidney lysates from WT and RC/RC mice ($n = 6$ per group). Graphs depict cumulative quantification of data; ($n = 6$ per group); $^*p < 0.05$.

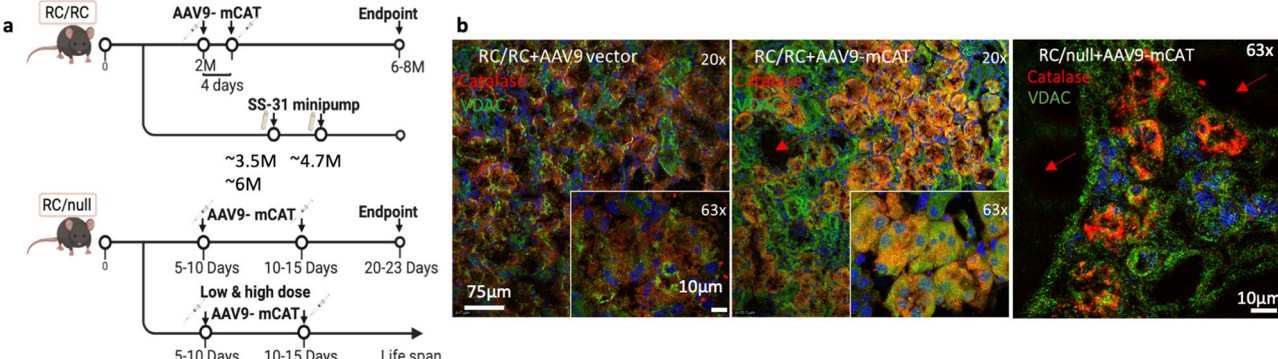

**Fig. 2 Strategies to protect mitochondria. a** Experimental design of the in vivo study (created by biorender). **b** Immunofluorescence of catalase (red) and the mitochondrial marker VDAC (green) in kidney tubular epithelial cells from RC/RC mice injected with AAV9 alone (RC/RC + AAV9 vector) or RC/RC and RC/null mice injected with AAV9 overexpressing catalase targeted to mitochondria (RC/RC + AAV9-mCAT) at low (20×) and high objective (63×) magnification (inset); $n = 3$ per group. Red arrows indicate cysts.

Hemoglobin levels were significantly decreased from 15.2 g/dL in WT to 12.13 g/dL in RC/RC mice, suggesting mild CKD-associated anemia, which was non-significantly mitigated by AAV9-mCAT (Fig. 3d). To further investigate the effect of mCAT treatment on cyst progression in kidneys of PKD1 mutant mice, we performed a pathological examination. Low power images showed multiple obvious cysts in RC/RC kidneys (Fig. 3e). Trichrome staining showed a significant increase in fibrosis in kidneys from RC/RC mice compared with WT kidneys, which was also attenuated by mCAT (Fig. 3f, g). Quantitative analysis of cyst area showed a significant increase in RC/RC mice compared to WT mice, which was significantly reduced upon treatment with AAV9-mCAT (Fig. 3h). We observed increased fibrosis in RC/RC kidneys at higher magnification, which was reduced upon treatment with mCAT (Fig. 3i). Analysis of abnormal areas replaced by either cyst or fibrosis shows ~30−40% of abnormal areas in RC/RC and treatment with AAV9-mCAT significantly reduced the combined cystic and fibrotic areas in RC/RC mice (Fig. 3j).

We observed greatly enlarged kidneys that were almost entirely replaced by cysts and fibrosis in ~24 day-old RC/null mice, as previously reported[17]. This enlargement was partially ameliorated in mice that received a low dose injection of AAV9-mCAT (Fig. 4a). Kidney weight normalized to body weight was increased by ~20-fold in RC/null mice compared with RC/WT littermate controls, which harbor only one PKD1 allele with the R to C point mutation and do not show any signs of kidney anomalies at this age. Normalized kidney weight in RC/null mice was significantly reduced by overexpression of mCAT ($p = 0.02$, Fig. 4b, Supplementary Fig. 2a). Serum BUN was >100 mg/dL in RC/null mice around weaning age, indicating uremia (Fig. 4c and Supplementary Fig. 2b). Furthermore, severe anemia (median Hb ~5 g/dL) was observed in nearly all RC/null mice injected with AAV9 vector (untreated group) (Fig. 4d and Supplementary Fig. 2b). Overexpression of mCAT significantly reduced the elevated serum BUN and attenuated the severe anemia, although many of the mCAT-treated mice still had serum BUN > 100 mg/

dL, suggesting the progression of renal failure was delayed but not prevented (Fig. 4c, d, and Supplementary Fig. 2b). Treatment with AAV9-vector or AAV9-mCAT did not significantly alter the body weight of RC/WT, RC/null, or RC/RC mice (Supplementary Fig. 2c). Our pathological examination showed excessively enlarged cysts in RC/null kidneys with little remaining kidney tissue, which was attenuated upon treatment with mCAT (Fig. 4e). Since most of the kidney parenchyma in RC/null mice was replaced by cysts, there was no further increase in fibrosis than that observed in RC/RC mice (Fig. 4f, g, i). RC/null mice had significantly increased cyst area compared to WT and RC/RC mice, which was also attenuated by mCAT (Fig. 4h). Our analysis showed ~60−80% of abnormal areas in RC/null mice replaced by either cyst or fibrosis (Fig. 4j). Treatment with AAV9-mCAT significantly reduced the combined cystic and fibrotic area in these mice (Fig. 4j).

The median survival of RC/null mice was ~20 days, with a maximal survival of ~30 days (Fig. 4k). Consistent with delaying the progression of PKD, mCAT significantly extended predicted median survival and maximal survival by 56% and greater than 2.5-fold, respectively (log-rank $p < 0.001$) (Fig. 4k). The hazard ratio of mCAT injection was 0.51 (95% CI: 0.29, 0.89; $p = 0.017$). To test the dose-effect relationship of mCAT treatment, we injected a subset of RC/null mice with a 5× higher dose of AAV9-mCAT ($10^{10}$/g), which showed a similar effect on survival as those that received a low dose ($2 \times 10^9$/g) (Supplementary Fig. 3).

**Targeted proteome profiling shows derangement of mitochondrial and metabolic pathways in PKD1 mutant mice, which are partially protected by mCAT.** To gain a deeper understanding of the potential molecular mechanisms driving the pathology and dysfunction of kidneys in PKD1 mouse models, we performed targeted proteomics profiling of several key mitochondrial and metabolic pathways using kidney tissue from RC/RC and RC/null mice that were untreated or treated with AAV9-

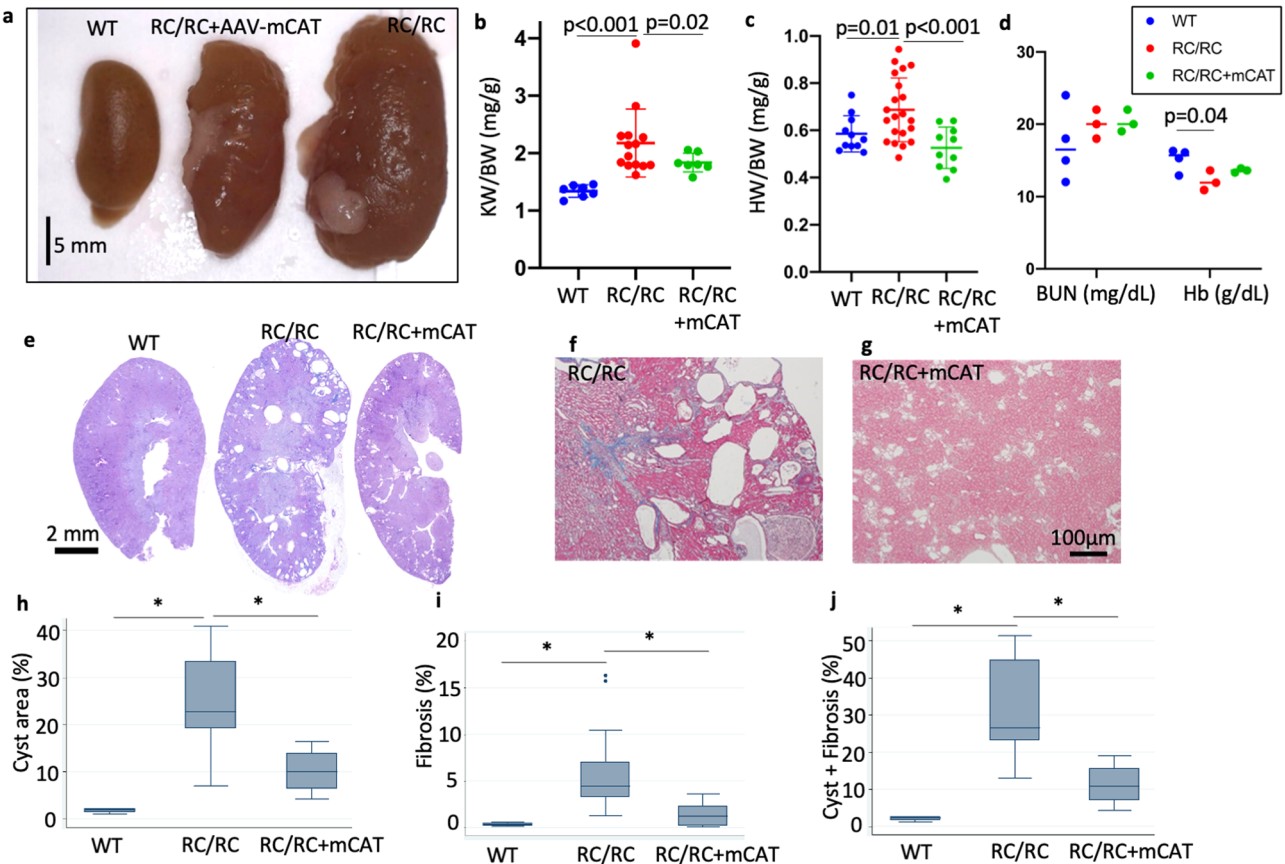

**Fig. 3 Overexpression of mCAT ameliorates kidney pathology, cystic enlargement, and fibrosis in RC/RC mice. a** Gross pathology of kidneys from 6−7-month-old WT and RC/RC mice or RC/RC mice that received low dose injection (2 × 10⁹/g) of AAV9-mCAT (RC/RC + AAV9-mCAT) (representative image shown; $n = 4−6$). **b** Kidney weight and **c** heart weight normalized to body weight in mice of the indicated genotypes and treatment groups; $n = 10−21$. **d** Blood urea nitrogen (BUN; mg/dL) and Hemoglobin (g/dL) in mice of the indicated genotypes and treatment groups; $n = 3−4$ per group. **e** Representative PAS staining of kidney sections from WT, RC/RC, and RC/RC treated with AAV9-mCAT. **f, g** Trichrome staining of kidney section from RC/RC and RC/RC treated with AAV9-mCAT. Quantitative analysis of **h** cysts area, **i** fibrotic area, and **j** combined cysts and fibrotic area. *$p < 0.05$.; ($n = 4$ for WT, $n = 15−18$ for each of the other groups).

mCAT. We assessed both mouse models, RC/RC exhibiting slowly progressive disease and RC/null exhibiting rapidly progressive PKD[9,10]. This gene dosage effect of PKD1 has been reported in humans with ADPKD and mouse models of the disease[18] and is supported by the more severe phenotype observed in 20-day-old RC/null mice compared with the phenotype observed in 6−7-month-old RC/RC mice (Figs. 3, 4). Using penalized multiple linear regression models, we analyzed the various effects that arise due to differences in PKD1 protein abundance, including the effects due to the PKD1 mutation, the gene dosage effects of the PKD1 mutation, and the effect of mCAT treatment in both RC/RC and RC/null mice.

Proteomics profiling of 137 mitochondrial and metabolic proteins showed 98 proteins were significantly altered between mice with PKD1 mutation (combined RC/RC and RC/null) relative to WT samples (Fig. 5a). Of these, 94 proteins were significantly suppressed in mice with a PKD1 gene mutation (Fig. 5a and Supplementary Data 1). In comparing the PKD1 gene dosage effect, we found that 119 proteins have more prominent changes in RC/null mice compared with RC/RC mice (Fig. 5a and Supplementary Data 1). Notably, treatment with mCAT significantly ameliorated the effects of PKD1 mutation in 65 proteins, with the expression of 56 of these being significantly reversed by greater than 10% (Fig. 5a and Supplementary Data 1). These data confirmed catalase was overexpressed by ~64% in mice that received AAV9-mCAT treatment (red asterisk, Fig. 5a

and Supplementary Data 1). Additionally, mCAT also restored expression of 16 of the 22 proteins involved in fatty acid oxidation (FAO) and 16 of the 26 proteins involved in the tricarboxylic acid (TCA) cycle and related pathways, which were altered due to PKD1 mutation.

Overall, PKD1 mutation altered proteins involved in many mitochondrial and metabolic pathways, and mCAT partially restored protein expression in several of these pathways. The average magnitude of the effect of PKD1 mutation and mCAT treatment on each pathway is summarized in Fig. 5b. Specifically, the top three pathways suppressed by the PKD1 mutation were the TCA cycle and related proteins (−0.31 or decreased by 31%), fatty acid beta-oxidation (FAO; −0.29), and respiratory complexes and related proteins (−0.28). Of these, mCAT had the strongest protective effect on FAO (80.7% restoration of PKD1 suppression). The proteins involved in FAO that had the greatest differences in expression upon mutated PKD1 were CPT1 & 2 and ACAD, all of which are key regulators of this process. PKD1 mutation also suppressed most proteins involved in the TCA cycle and endogenous antioxidants, many of which were significantly ameliorated by mCAT (Fig. 5a, b and Supplementary Data 1). Consistent with strong suppression of mitochondrial electron transport in RC/null kidneys (Fig. 1a–e), decreased levels of respiratory complexes were observed in both PKD1 mutant mice. The protective effects of mCAT were significant for succinate dehydrogenase subunits A&B (but not

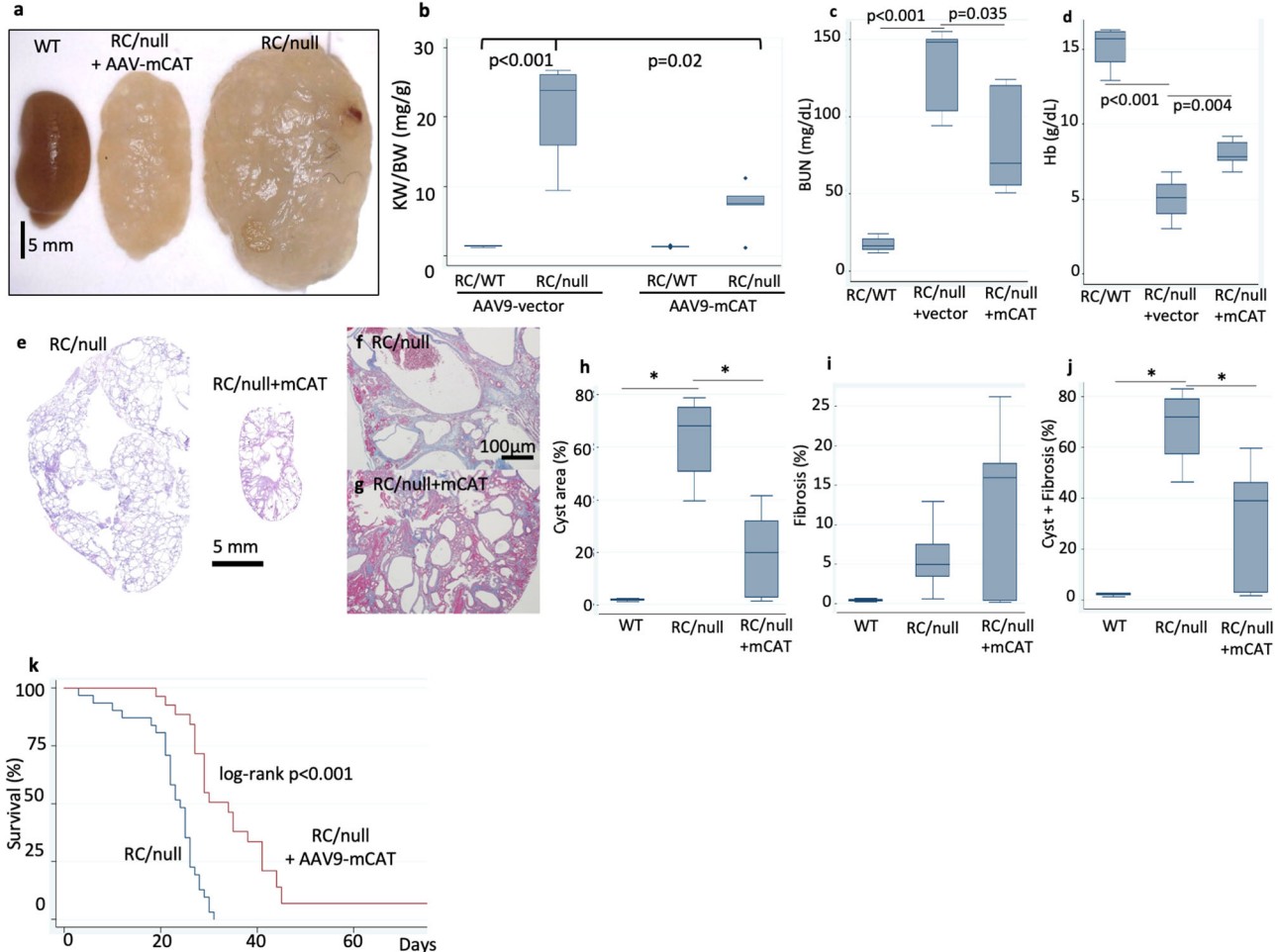

**Fig. 4 Overexpression of mCAT delays renal failure and mortality in RC/null mice. a** Huge RC/null gross pathology of kidneys from 24-days-old WT and RC/null mice, and RC/null mice treated with AAV9-mCAT ($2 \times 10^9$/g) (representative image shown; $n = 4-6$). **b** Kidney weight normalized to body weight in RC/WT or RC/null mice treated with AAV9 vector alone (AAV9-vector) or AAV9-mCAT. **c** BUN (mg/dL) and **d** hemoglobin (g/dL) in RC/WT mice or RC/null mice treated with AAV9 vector alone (AAV9-vector) or AAV9-mCAT; $n = 3-7$. **e** Representative PAS staining of kidney sections from RC/null mice, untreated and treated with AAV9-mCAT. **f**−**g** Trichrome staining of kidney section from RC/null mice untreated and treated with AAV9-mCAT. **h**−**j** Quantitative analysis of cysts area (**g**), fibrotic area (**h**), and combined cysts and fibrotic area (**i**). *$p < 0.05$.; $n = 4$ for WT; $n = 15-18$ for each of the other groups. **k** Survival curves of RC/null mice untreated or treated with AAV9-mCAT ($2*10^9$ /g); $n = 31$ for RC/null, $n = 19$ for RC/null+AAV9-mCAT.

subunit C) and electron transfer flavoproteins (etfa, etfb, and etfdh). However, mCAT did not significantly reverse the decreased levels of complex I and III subunits associated with the PKD1 mutation (Fig. 5a and Supplementary Data 1).

To determine the functional impact of mCAT treatment, we measured mitochondrial complex activity in kidney lysates from RC/null. Consistent with our proteomics data, AAV9-mCAT did not restore the substantially decreased activity of complexes I−IV (Supplementary Fig. 4a, b), neither did it restore the citrate synthase (mitochondrial mass) in RC/null kidney lysates (Supplementary Fig. 4c). However, AAV9-mCAT ameliorated the decrease in citrate synthase in RC/RC kidneys, with borderline significance ($p = 0.097$, Supplementary Fig. 4d). Collectively, these data indicate that PKD1 mutations disrupt the expression of proteins involved in multiple mitochondrial and metabolic processes, and that a subset of these can be restored by overexpression of catalase targeted to mitochondria.

**Scavenging mitochondrial ROS in RC/RC mice attenuates ERK1/2 phosphorylation and enhances AMPK phosphorylation.** To further investigate the signaling mechanisms related to increased mitochondrial ROS upon PKD1mutation, we utilized

immortalized human cyst-derived cells with a heterozygous PKD1 mutation (WT 9-7) and a proximal tubular cell line derived from a normal human kidney (HK-2, designated as control cells). These PKD1 mutant cells (WT9-7) had ~26% higher cell counts than control HK-2 cells ($p = 0.0021$, Fig. 6a). To elucidate the role of mitochondrial ROS in the signaling of PKD1 mutant cells, we treated the cells for 24 h with Mito-Tempo (Mtmp, 25 μM), a scavenger of mitochondrial superoxide, or mitoparaquat (MPQ, 10 μM), an inducer of mitochondrial superoxide, or AAV9-mCAT (10 μM) or SS31 (10 μM). SS31 is a tetrapeptide that has been shown to protect mitochondria through interaction with cardiolipin, enhancement of ATP synthesis, and reduction of mitochondrial ROS[19]. Figure 6a shows cell counts 24 h after treatment. Saline-treated PKD1 mutant cells have significantly higher cell counts than control cells. MPQ treatment of PKD1 cells substantially increased the number of cells ($p < 0.001$). In contrast, Mtmp, mCAT, or SS31 decreased the cell count of PKD1 mutant cells, although only mCAT reached statistical significance and the effect of SS31 was borderline ($p = 0.002$ and 0.155, respectively, Fig. 6a). Next, to demonstrate whether alteration in mitochondrial ROS affects proliferation rate, we performed immunostaining of Ki-67 in

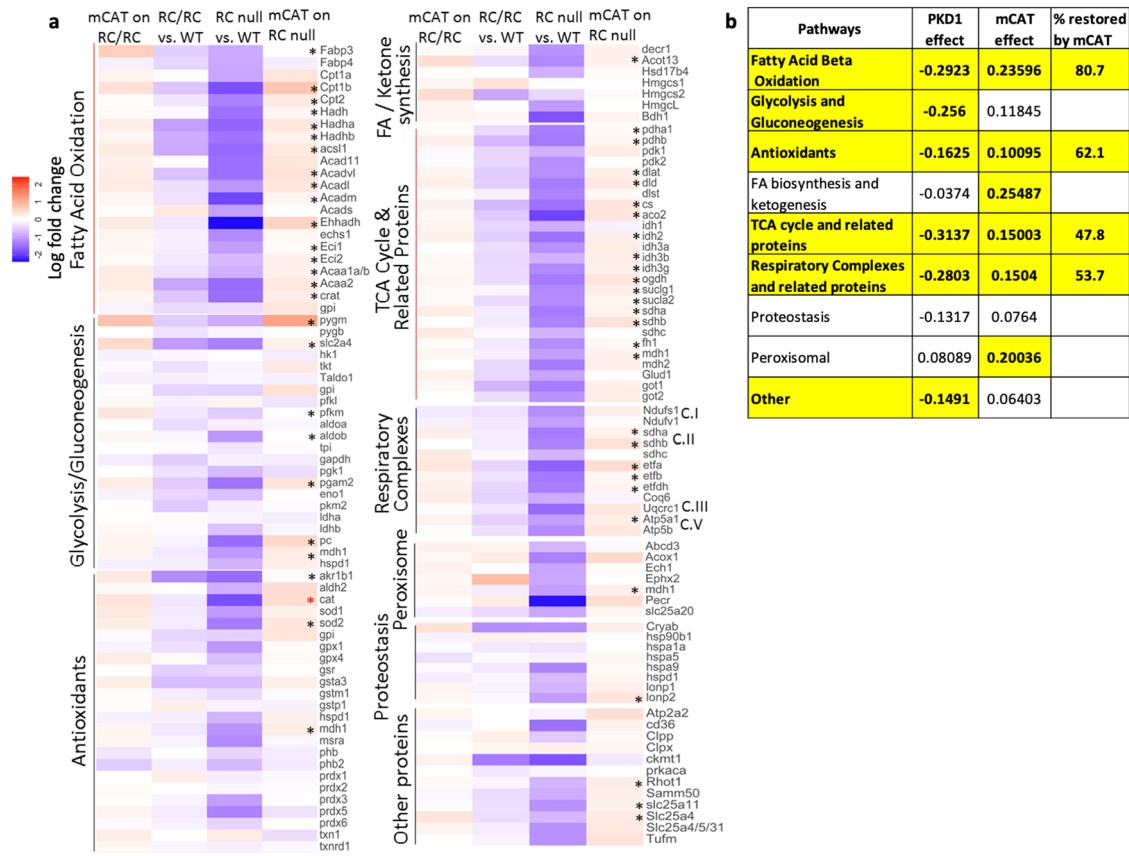

| Pathways | PKD1 effect | mCAT effect | % restored by mCAT |
|---|---|---|---|
| Fatty Acid Beta Oxidation | -0.2923 | 0.23596 | 80.7 |
| Glycolysis and Gluconeogenesis | -0.256 | 0.11845 | |
| Antioxidants | -0.1625 | 0.10095 | 62.1 |
| FA biosynthesis and ketogenesis | -0.0374 | 0.25487 | |
| TCA cycle and related proteins | -0.3137 | 0.15003 | 47.8 |
| Respiratory Complexes and related proteins | -0.2803 | 0.1504 | 53.7 |
| Proteostasis | -0.1317 | 0.0764 | |
| Peroxisomal | 0.08089 | 0.20036 | |
| Other | -0.1491 | 0.06403 | |

**Fig. 5 Targeted proteomics profiling of mitochondrial and metabolic pathways. a** Heat maps of relative protein abundance from kidneys isolated from both PKD1 mouse models, RC/RC and RC/null, vs. WT controls in the middle columns and the mCAT treatment effect next to the corresponding group. The middle columns showed that most proteins were in blue, indicating a decreased abundance of several proteins in PKD1 mutation. The mCAT effects were mostly in red, suggesting various degrees of reversal of the PKD1 mutation effect. The asterisks highlight the proteins that show significant suppression by PKD1, with significant gene dosage effect and significant reversal by mCAT. (9 RC/RC, 7RC/RC treated with mCAT, 4 RC/null, 3 RC/null treated with mCAT, and 4 WT mice were used) **b** Table showing the average magnitude of the effects of PKD1 mutation, mCAT treatment, and the percentage restored by mCAT treatment on proteins involved in mitochondrial and metabolic pathways, as identified in A. Only highlighted numbers are significant.

PKD1 mutant cells. We showed that treatment with MPQ for 24 h significantly increased the proportion of Ki-67 positive cells, whereas treatment with Mtmp for 24 h significantly decreased the proportion of Ki67 positive cells compared with saline-treated PKD1 cells (Fig. 6b). These suggest that mitochondrial ROS induce cellular proliferation. Related to this, we performed IHC staining of Ki-67 in RC/RC and RC/null kidney tissues and showed that Ki-67 staining was increased in RC/RC and RC/null mice kidneys. However, this increase was not significant (Supplementary Fig. 5) since the Ki-67 labeling was much lower in vivo compared with in vitro PKD1 cells. In PKD1 mutant cells, MPQ decreased AMPK phosphorylation and induced ERK phosphorylation, whereas treatment with Mtmp had the opposite effect, increased AMPK phosphorylation and decreased ERK phosphorylation (Fig. 6c–f and Supplementary Figs. 8, 9). However, there were no significant changes in AMPK and ERK phosphorylation in control HK2 cells in response to these treatments (Supplementary Fig. 6a, b). Similar to MPQ treated PKD1 cells, we observed increased ERK phosphorylation in RC/RC mouse kidneys compared to WT kidneys, and this was significantly prevented in RC/RC mice treated with mCAT (Fig. 6g and Supplementary Fig. 10). AAV9-mCAT treatment also non-significantly increased AMPK phosphorylation in vivo (Supplementary Fig. 6c). These findings suggest the role of mitochondrial ROS in stimulating cell proliferation via ERK phosphorylation

and the inhibitory effect of mitochondrial ROS on AMPK, one of the master regulators of cellular metabolism.

**SS31 attenuates kidney cysts progression in RC/RC mice.** To investigate the in vivo effect of SS31, we inserted two sequential subcutaneous Alzet 1004 minipumps loaded with SS31 (3 µg/g/d) into RC/RC mice to deliver the drugs for 8−10 weeks continuously, starting at 3.5 months until mice were 6 months old (Fig. 2a). Treatment with SS31 significantly reduced kidney size and normalized kidney weight in RC/RC mice (Fig. 7a, b). PAS staining of kidney sections showed that SS31 significantly reduced kidney cysts areas (Fig. 7c, d). Furthermore, trichrome staining showed that SS31 also significantly decreased fibrosis in RC/RC mice (Fig. 7e–g). These findings suggest a potential benefit of SS31 for the treatment of ADPKD patients.

**Increased ROS and oxidative damage in RC/RC kidneys mitigated by mCAT or SS31.** Previous publications and our data above indicate the critical role of ROS-induced mitochondrial damage in the context of mutated PKD1[20]. To measure the relative abundance of ROS, we performed ex-vivo live staining of ROS in fresh kidney sections[21] from RC/RC mice. It revealed significant increases in the levels of mitochondrial superoxide (Mitosox) and total cellular ROS (DCFDA), which were

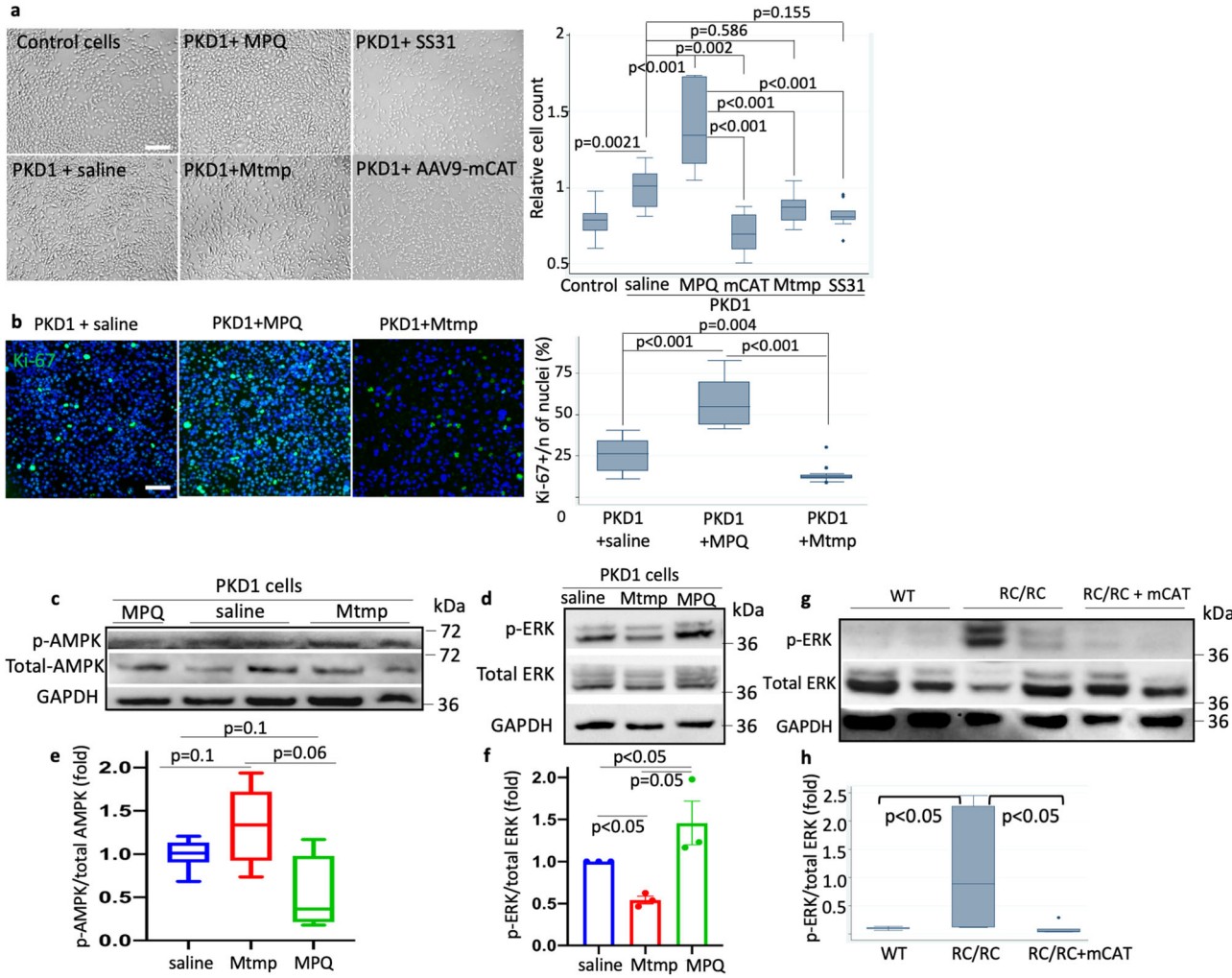

**Fig. 6 Increased ERK1/2 phosphorylation and decreased AMPK phosphorylation in PKD1 mutant cells and mouse model and the effect of mitochondrial ROS. a** Representative images of HK-2 control cells and WT9-7 PKD1 mutant cells treated for 24 h with saline, 25 µM Mito-Tempo (Mtmp), 10 µM mito-paraquat (MPQ), 10 µM AAV9-mCAT or 10 µM SS31 and the cell count (scale bar: 100 µm) ($n = 3$ per group). **b** Immunostaining of Ki-67 in PKD1 cells treated with saline, Mtmp, or MPQ and quantification of proliferation marker Ki-67; images are representatives of three independent experiments (scale bar: 100 µm). The effect of MPQ and Mtmp on the phosphorylation of AMPK and ERK shown by immunoblotting of **c** phospho and total AMPK and **d** phospho and total ERK in PKD1 mutant cells (WT9-7) and **e, f** quantification of indicated groups. GAPDH was used as loading control; $n = 3$. **g** phospho and total ERK in WT, RC/RC, and RC/RC treated with AAV9-mCAT, **h** quantification of indicated groups. $n = 3$ for controls; $n = 6-9$ for treatment groups.

attenuated upon treatment with AAV9-mCAT (Fig. 8a–c, e). SS31 decreased ROS in RC/RC kidneys, similar to that observed with mCAT (Fig. 8d, e). In addition, in RC/RC and RC/null mice, treatment with AAV9-mCAT significantly decreased the elevated levels of F2 isoprostane, an oxidative marker produced by ROS-induced peroxidation of arachidonic acid. A similar effect was seen with SS31 treatment in RC/RC mice (Fig. 8f). To investigate the extent of tissue oxidative damage, we performed IHC staining for nitrotyrosine (NT), a marker of peroxynitrite formation. We analyzed NT staining intensity in proximal and distal tubules as well as epithelial cells lining the enlarged cysts. There was a ~70% increase in NT staining intensity in both proximal and distal tubules of RC/RC kidneys compared with WT controls ($p < 0.001$ for both). While there was no significant difference in NT staining between proximal and distal tubules ($p = 0.369$), there was significantly higher NT staining in cystic epithelial cells than in proximal or distal tubules of RC/RC kidneys ($p < 0.001$, Fig. 8h–k), indicating higher oxidative damages in cystic tubules. Treatment with AAV9-mCAT decreased NT staining in both proximal and distal tubules of RC/RC kidneys, although this was

only significant in the proximal tubules ($p < 0.001$) but not distal tubules ($p = 0.12$), consistent with the fact that proximal tubules have more mitochondria. NT staining was slightly attenuated in the cystic epithelial lining in RC/RC kidneys treated with mCAT ($p = 0.08$) (Fig. 8j, k). Since RC/null kidneys were mainly consisted of cysts and fibrotic tissue, comparison between RC/null and RC/null treated with mCAT was not feasible (Supplementary Fig. 7).

## Discussion

In this study, we demonstrate a direct role for mitochondrial oxidative stress in both slowly progressive (RC/RC) and rapidly progressive (RC/null) mouse models of ADPKD. In brief, our study highlights five notable findings: (1) mitochondrial-targeted catalase (mCAT) delivered by AAV9 significantly ameliorated the phenotypes of both PKD1 mouse models that were studied, in parallel with reducing mitochondrial ROS and tissue oxidative damage. (2) Targeted proteomics studies using our PKD1 mouse models highlighted that proteins involved in FAO, TCA cycle,

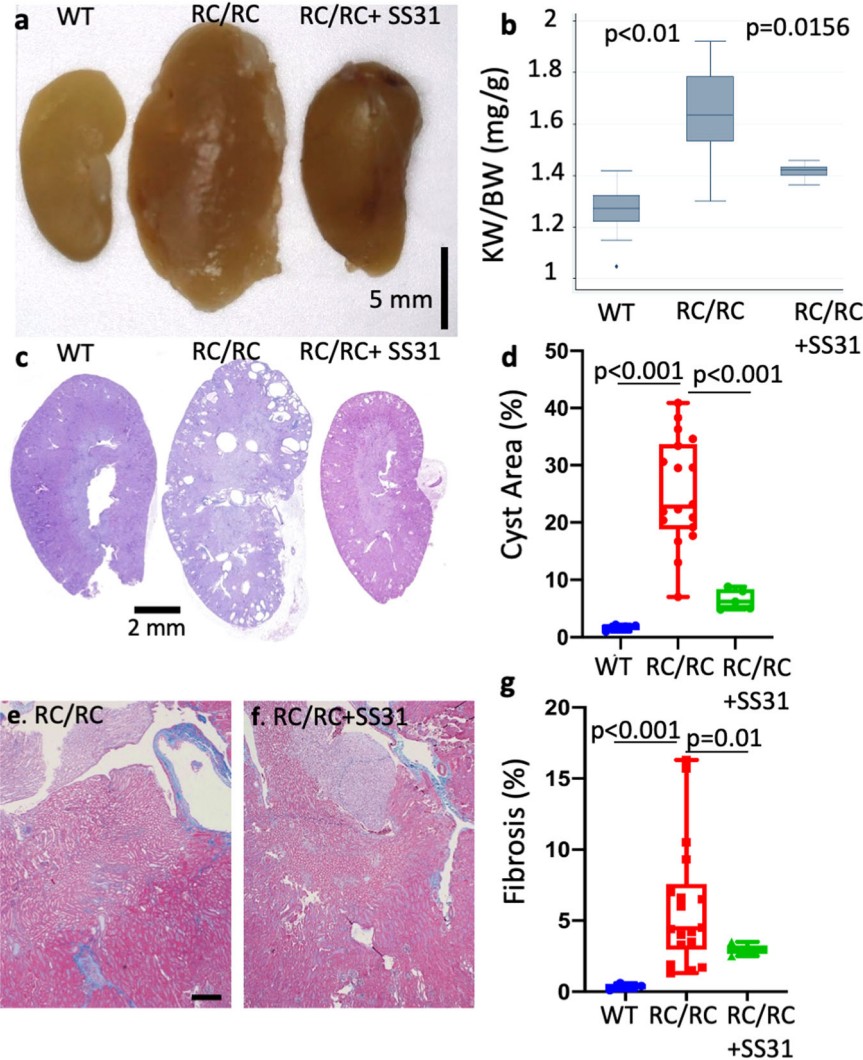

**Fig. 7 SS31 exerts a protective effect in RC/RC kidneys. a** Gross pathology **b** normalized kidney weight from WT, RC/RC, and RC/RC mice treated with SS31. **c** Representative PAS staining of kidney sections. **d** Quantitative analysis of cysts area. **e**, **f** Higher magnification of trichrome staining of kidney sections from RC/RC and RC/RC mice treated with SS31 (scale bar: 100 μm). **g** Quantitative analysis of trichrome blue area; $n = 5-10$.

respiratory complexes, glucose utilization, and cellular anti-oxidants were significantly suppressed by PKD1 mutations. (3) Injection with AAV9-mCAT significantly restored, at least in part, the levels of proteins involved in FAO, endogenous anti-oxidants, TCA cycle, and respiratory complexes and related proteins. (4) Mitochondrial ROS induced ERK1/2 phosphoryla-tion and inactivated/dephosphorylated AMPK in human PKD mutant cell lines, whereas scavenging mitochondrial ROS atte-nuated ERK1/2 phosphorylation and increased AMPK phos-phorylation. (5) SS31 mitochondrial protective tetrapeptide treatment mitigated the progression of ADPKD-like disease, in parallel with reducing mitochondrial ROS and oxidative damage, similar to the effects of mCAT.

According to the widely accepted two-hit hypothesis for PKD, a second insult in addition to germline homozygous PKD mutations is required for renal cystogenesis in ADPKD[17,22,23]. One such potential second hit is oxidative stress, which has been shown early in the course of disease in ADPKD patients and murine models[24], long before their renal function is impaired[16,18]. A recent study using proteomics and tran-scriptomics approach reported that NRF2-regulated antioxidant pathways were suppressed in ADPKD, while the activation of NRF2 ameliorated oxidative stress and cystogenesis in ADPKD[25].

This suggests that oxidative stress plays a crucial role in cyst formation/progression[23,26–28].

Mitochondria are major sites of intracellular ROS generation. ROS as signaling molecules may impact metabolism and cellular proliferation. Since tubular epithelial cells are highly enriched in mitochondria, we investigated the role of mitochondrial ROS in cystogenesis and the progression of PKD. Mitochondrial ROS has been implicated in aging and multiple diseases[29]. To investigate the role of oxidative stress in different subcellular compartments, Schriner et al. generated mice overexpressing catalase targeted to mitochondria (mCAT) or the natural sites within peroxisomes (pCAT). The beneficial effect of mCAT exceeded the effect of pCAT overexpression in terms of murine lifespan extension[15], cardiac hypertrophy, and failure[12]. These findings emphasize the critical roles of targeting mitochondrial oxidative stress. Since then, mCAT has been shown to ameliorate various age-related diseases and cancers[29].

Our targeted proteomics data demonstrated that there was, on average, an ~16% decrease in the abundance of most antioxidant enzyme systems in kidneys from mice with PKD1 mutation, including superoxide dismutases (SOD) I and II, catalase, glu-tathione antioxidant system, and the majority of peroxiredoxins (Fig. 5A, B and Supplementary Data 1). Furthermore, we

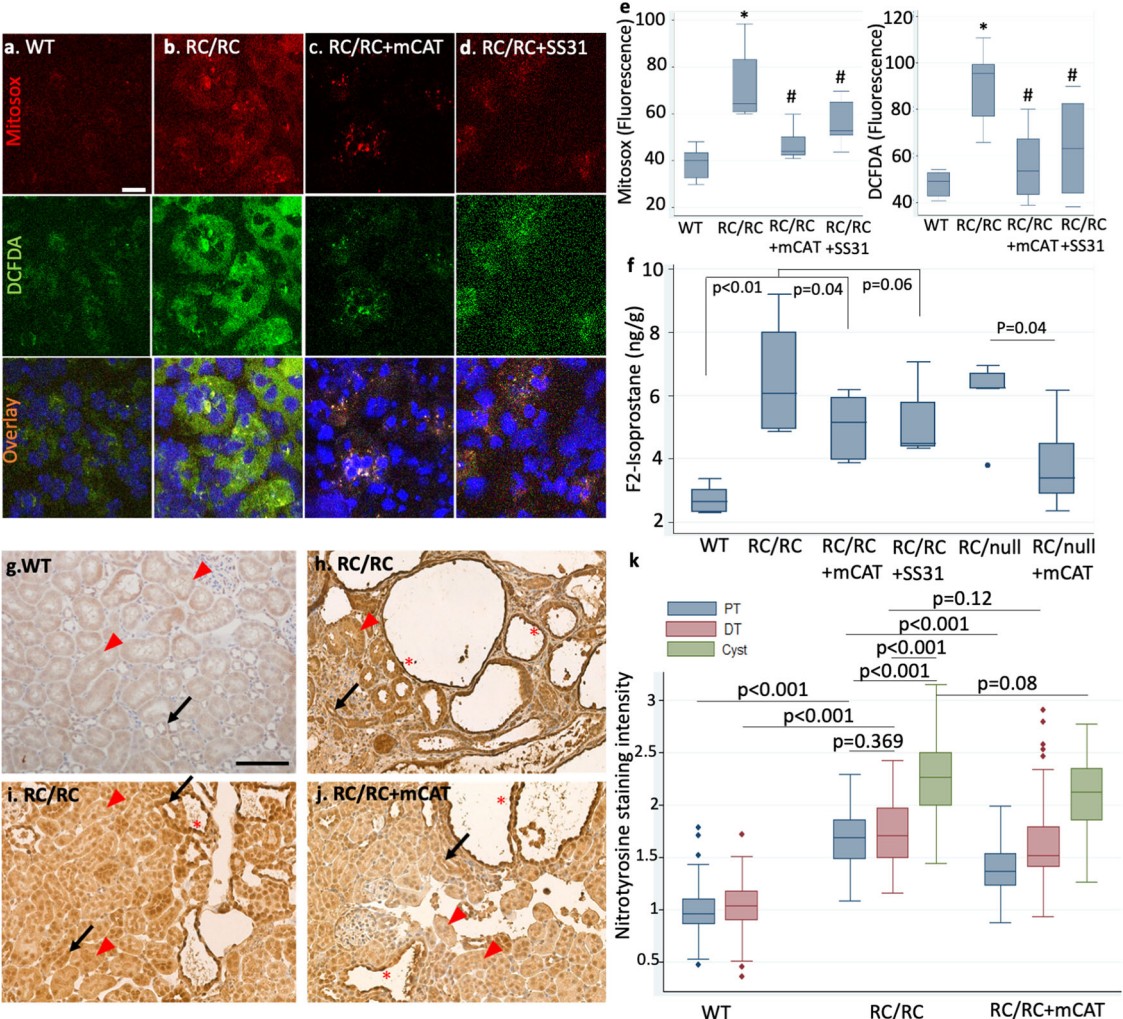

**Fig. 8 ROS and oxidative damage in RC/RC kidneys.** Representative images of ex vivo live staining of kidney slices with MitoSOX (top row), an indicator of mitochondrial superoxide and DCFDA (middle row), an indicator of total cellular ROS, overlaid with Hoechst 33342 (bottom row) in kidneys from **a** WT and **b** RC/RC mice or RC/RC mice treated with **c** low dose AAV9-mCAT (RC/RC + mCAT) or **d** the mitochondrial protective peptide SS31; $n = 3-4$ per group (scale bar: 25 µm). **e** Quantification of ROS from cells treated as in (**a–d**). **f** Concentration of F2-isoprostane, a marker of lipid peroxidation in kidney tissue of WT, RC/RC, RC/RC treated with mCAT or SS31, RC/null and RC/null treated with mCAT, measured by high-performance liquid chromatography; $n = 4-6$. *$p < 0.01$ vs. WT; #$p < 0.05$ vs. RC/RC. IHC staining for nitrotyrosine in **g** WT, **h**, **i** RC/RC, and **j** RC/RC treated with AAV9-mCAT and **k** quantification of nitrotyrosine staining intensity as in (**g–j**). (Scale bar: 100 µm); PT: red arrowhead, DT: black arrow, *: cysts; $n = 3$ per group; PT: proximal tubule, DT: distal tubule.

observed increased mitochondrial and total cellular ROS in kidney sections of RC/RC mice and a consistent increase in oxidative damage (Fig. 8e, f). Notably, treatment with mCAT significantly decreased mitochondrial and total cellular ROS and increased mitochondrial SOD II (Fig. 5a), ameliorated oxidative damage, cyst progression, and PKD-associated cardiac hypertrophy in the slowly progressive model of ADPKD (i.e., RC/RC mice; Fig. 3). In the rapidly progressive model of PKD, RC/null mice, over-expressing mitochondrial-targeted catalase mitigated cyst progression, the development of uremia, and severe anemia. In addition, this treatment extended the median and maximal survival of these mice (Fig. 4). A higher dose of AAV9-mCAT did not provide additional survival benefits to RC/null mice (Supplementary Fig. 3). This finding is consistent with our previous study showing that the mitochondrial targeting of catalase itself is more crucial to its protective effect than the magnitude of its overexpression[29]. In this current study, a low dose of AAV9-mCAT resulted in an ~60% increase in the overall abundance of catalase (Fig. 5a), which was sufficient to confer a mitochondrial

protective effect. Indeed, the concept of protecting mitochondria in PKD1 is in parallel with a recent study reporting that increased fragmentation and impaired mitochondrial function play a role in PKD1 progression[30].

In this study, we applied a state-of-the-art targeted proteomics approach to measure the abundance of 137 proteins involved in several mitochondrial and metabolic pathways, as well as those involved in the peroxisomal and proteostasis pathways. We demonstrated that the top three pathways significantly suppressed by the effect of PKD1 mutations were the TCA cycle (−31%), FAO (−29%), and respiratory complexes (−28%). The latter was confirmed by substantial decreases in mitochondrial respiratory complex activity (Fig. 1a). Interestingly, the pattern of proteome changes in the current study is reinforced by a comprehensive metabolomics study, which reported that the most striking changes of metabolites in PKD1 suggested a profound impairment of the TCA cycle, FAO, and fatty acid synthesis, glycolysis, as well as pentose phosphate pathway[31]. We showed that AAV9-mCAT partially restored the decreased levels of

proteins in FAO (80.7%), antioxidant systems (62.1%), TCA cycle (47.8%), and respiratory complexes and related proteins (53.7%). However, partial preservation of some respiratory complex subunits by mCAT apparently did not improve the overall respiratory complex activities, as measured by extracellular flux analysis (Supplementary Fig. 4a, b). This may be due to the fact that some key mitochondrial encoded subunits remain present at suboptimal levels. It is important to note that the mitochondrial electron transport activity measured here is likely far in excess of that needed to maintain ATP production. In addition, the activity measured in frozen tissue (Fig. 1) relies on exogenous substrates, which could be limited by the PKD1 mutation. This may explain the observation that mCAT improved the progression of PKD and the levels of many mitochondrial metabolic enzymes but does not enhance electron transport activity. Furthermore, although the positive correlation between loss of mitochondrial mass and the progression of cysts in two models of PKD1 does not provide evidence of causality, we propose that there is a vicious cycle between mitochondrial dysfunction and increased ROS leading to the progression of cysts, which may subsequently induce a further decrease in mitochondrial function. The replacement of tubules by cysts and fibrosis may also contribute to further reduction of mitochondrial mass and progressive decline in mitochondrial function. The fact that scavenging mitochondrial ROS by mCAT ameliorated mitochondrial and metabolic proteome, attenuated cyst progression, and prevented the decrease in mitochondrial mass in RC/RC kidneys (borderline significance) suggests breaking the vicious cycle may delay PKD progression.

Since ERK1/2 has been shown to be a ROS-sensitive kinase[32], we demonstrate that mitochondrial ROS generated by mitoparaquat (MPQ) induced ERK1/2 phosphorylation in human PKD1 immortalized cell lines (WT9-7), whereas scavenging ROS by MitoTempo significantly mitigated ERK1/2 phosphorylation (Fig. 6d). ERK1/2 belongs to the mitogen-activated protein kinase superfamily that controls cell proliferation. In addition, ERK1/2 inhibits the liver kinase B1 (LKB1)-AMP-activated protein kinase (AMPK) axis[33]. AMPK may be directly regulated by ROS, such as by oxidation of cysteine residues in its α subunit, which interferes with the activation/phosphorylation of AMPK[34]. AMPK can modulate FAO by two mechanisms: through phosphorylation or inactivation of acyl CoA carboxylase (ACC) and activation of PPARα. AMPK inhibits ACC and reduces malonyl CoA, an inhibitor of carnitine palmitoyl transferase-1 (CPT-1), the first rate-limiting step of FAO (Fig. 9). Hence, reducing mitochondrial ROS results in two different consequences: (1) mitigating ERK1/2 phosphorylation and thereby reducing tubular cell proliferation, and (2) restoring the FAO and TCA cycle through preservation of AMPK activity (Fig. 9). It is noteworthy that renal tubular epithelial cells are rich in mitochondria and utilize fatty acid as preferred substrates to generate higher amounts of ATP to carry out various forms of energy-demanding active transport. Thus, reducing cell proliferation and preserving the metabolic pathway markedly ameliorates PKD progression (Fig. 9).

For potential translation into clinical therapeutics, we tested the effect of the SS31 mitoprotective peptide in our mouse model of PKD1 mutation. SS31 is a tetrapeptide that binds to cardiolipin, enhances the efficiency of electron transfer and ATP production, and indirectly decreases ROS. It is currently in multiple phase II and phase III clinical trials for heart failure, renal ischemia-reperfusion injury, and mitochondrial myopathies[35–37]. The beneficial effects of SS31 on cyst progression and oxidative damage are comparable to the antioxidant effect of mCAT treatment in RC/RC mice (Figs. 6–8), although the pleiotropic effects of SS31 may involve additional mechanisms that warrant future investigation.

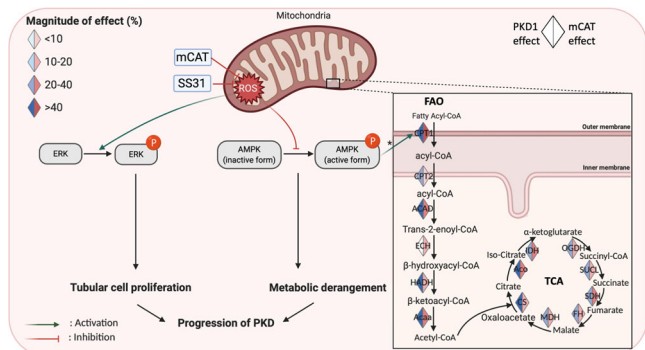

**Fig. 9 Schematic illustration of the molecular mechanisms downstream of ROS in PKD1 mutant kidneys.** Mitochondrial ROS induce activation/phosphorylation of ERK, inhibition of AMPK, and the subsequent decreases (indicated by blue) in most enzymes involved in the fatty acid oxidation (FAO) and tricarboxylic acid (TCA) cycle. These enzymes were increased (indicated by red) by mCAT treatment of PKD1 mutant mice, suggesting a partial restoration of FAO and TCA, * through inhibition of acyl-CoA carboxylase and reduction of malonyl-CoA, an inhibitor of FAO (created by biorender).

In summary, we demonstrate that protecting mitochondria by overexpressing mitochondrial-targeted catalase or SS31 ameliorates the progression of PKD1 in mouse models of the disease. The potential mechanisms include mitigation of ERK1/2 phosphorylation, preservation of AMPK, and partial reversal of metabolic derangement in PKD1. In addition, our findings support the potential clinical application of SS31 for the treatment of PKD1.

## Methods

**Mouse studies and experimental plan.** Animal experiments were approved by the University of Iowa Animal Care and Use Committee. *Pkd1* p.R3277C knock-in mice were obtained from Mayo Clinic[17]. They were the result of the *Pkd1* p.R3277C mutation introduced using the exact codon found in ADPKD probands (*Pkd1* c.9805.9807AGA > TGC) as described[17]. The PKD1$^{RC/null}$ and RC/WT control mice were generated by crossing PKD1$^{RC/RC}$ mice with PKD1$^{+/−}$ mice. The PKD1$^{+/−}$ were generated by breeding PKD1$^{flox/flox}$ mice (B6.129S4-*Pkd1*$^{tm2Ggg}$/J, JAX, 129S4/SvJae background) with germline Sox2-Cre transgenic mice (Edil3$^{Tg(Sox2-cre)1Amc}$, JAX, C57BL/6 background). All mice were fed with regular chow.

The experimental design is illustrated in Fig. 2a. Briefly, treatment with intraperitoneal injection of AAV-mCAT was initiated at 3−4 months-of-age for RC/RC, at 5−10 days for RC/null, and then repeated after 4 days, using the gene delivery vehicle, Adeno-associated virus (AAV) serotype 9 as described[38]. The mCAT construct was provided by Drs. Peter Rabinovitch (University of Washington) and Dongsheng Duan (University of Missouri). We performed continuous SS31 treatment using two sequential subcutaneous insertions of the Alzet 1004 minipump at ~3.5 month-of-age for RC/RC mice, then replaced by the second pump 5 weeks after the first pump.

**Measurement of BUN and hemoglobin.** At the end of each experiment, mice were euthanized, and blood was collected in a heparinized tube via retro-orbital bleeding. Fresh blood was tested using an iSTAT analyzer with CHEM8+ cartridges. Since serum creatinine was below the detection limit by iSTAT for most mice, it was not reported in the current study.

**Measurement of mitochondrial respiratory complex activity.** Kidney tissues from 9 WT (three of them 4−6 weeks old and six 4−6 months old), 3 RC/null (3 weeks old), 3 RC/null treated with mCAT, 6 RC/RC (6−8 months old), and 6 RC/RC treated with mCAT were flash-frozen in liquid nitrogen. Frozen kidney tissues were pulverized in a liquid nitrogen-cooled mini mortar and then mechanically homogenized with 10 strokes in a glass–glass Dounce homogenizer with MAS buffer (70 mM sucrose, 220 mM mannitol, 10 mM KH$_2$PO$_4$, 5 mM MgCl$_2$, 1 mM EGTA, 2 mM HEPES pH 7.2). All homogenates were centrifuged at 1,000×*g* for 10 min at 4 °C followed by the collection of the supernatant. Protein concentration was determined by a DC Lowry protein assay (BioRad).

The kidney homogenates (4 μg protein) were loaded into Seahorse XF96 microplates in 20 μl of MAS buffer. The loaded plate was centrifuged at 2,000×g for 20 min at 4 °C (no brake), then 160 μl of MAS containing cytochrome c (10 μM) and alamethicin (10 μg/ml) were added per well. To determine specific complex activity, the following concentration of substrates and inhibitors were used: Complex 1: NADH (1 mM) followed by rotenone (2 μM); Complex-II: 5 mM succinate + rotenone (5 mM + 2 μM), followed by antimycin A (20 μM); Complex-III: duroquinol (0.5 mM) followed by antimycin A (20 μM); Complex-IV: Tetamethylphenylenediamine (TMPD) + ascorbic acid (0.5 mM + 1 mM), followed by azide (50 mM)[13].

**Measurement of citrate synthase and lactate dehydrogenase (LDH) activity**. Citrate synthase activity was measured spectrophotometrically. Briefly, the change in absorbance of 5,5′-dithiobis(2,4-nitrobenzoic acid) (DTNB) was followed after the addition of kidney homogenates (15 μg) in the presence of 100 mM Tris/ 0.1% Triton X-100, (pH 8.0 at 37 °C), and 10 mM acetyl-CoA. Reactions were initiated by the addition of oxaloacetate (20 mM). Specific activity was calculated as the rate change normalized to total protein[39,40]. For measurement of LDH activity, homogenates (10 μg) were used to measure LDH activity spectrophotometrically by monitoring the rate of oxidation of NADH at 340 nm over time in the presence of 10 mM pyruvate. Reactions were performed at 37 °C in PBS, pH 7.4 containing 0.1% Triton x-100[41].

**Immunohistochemistry, immunofluorescence, and immunoblotting**. For immunohistochemistry and immunofluorescence, kidney slices were placed in OCT, and sections were processed with standard procedures. For immunoblotting, kidneys were homogenized in RIPA buffer containing protease inhibitor cocktail (Roche). After quantification with BCA, equal amount of proteins (30−100 μg) were loaded onto an SDS-PAGE gel, followed by standard immunoblotting. Band densities were quantified with Image J (NIH). Primary antibodies used were AMPKα (Cell Signaling Technology, 2532, 1:1000), phospho-AMPKα (Thr172) (Cell Signaling Technology, 2531, 1:1000), p44/42 MAPK (Erk1/2) (Cell Signaling Technology, 4695, 1:500) and phospho-p44/42 MAPK (ERK1/2) (Cell Signaling Technology, 4370, 1:500), anti-Catalase (R&D, AF3398, 1:50) and Ki-67 (abcam, ab16667, 1:200). Secondary antibodies were anti-rabbit IgG-HRP (Santa Cruz Biotechnology, sc-2357, 1:5000), anti-mouse IgG-HRP (Abcam; ab97046, 1:5000), Dolichos Biflorus Agglutinin (DBA) Fluorescein (Vector lab, FL-1031, 1:50), Lotus Tetragonolobus Lectin (LTL) Fluorescein (Vector labs, FL-1321-2, 1:50), donkey-anti-rabbit IgG Alexa Fluor 647 (Invitrogen, A-31573, 1:200), donkey-anti-goat IgG Alexa Fluor 555 (Invitrogen, A-21432, 1:200).

**Histological and cyst quantification analysis**. Kidney sections (2 μm) were stained with Masson trichrome to assess the degree of interstitial fibrosis. Low power images from each whole kidney section stained with PAS were obtained using a Leica slide scanner and used to quantify the relative areas of cysts and the area of blue staining (indicating fibrosis) relative to the total tissue area. The percentage of cystic areas and fibrotic areas were calculated using ImageJ (NIH).

**Measurement of oxidative damage**. F2-isoprostanes levels in kidneys were measured[42], with some modifications. Briefly, 100 mg of tissue were homogenized in 10 ml of ice-cold Folch solution (CHCl3: MeOH, 2:1). F2-isoprostanes were extracted and the internal standard, [2H4]8-Iso-PGF2α, was added to samples at the beginning of extraction to correct for yield of the extraction process, and levels were quantified by gas chromatography-mass spectrometry. Esterified F2-isoprostanes were measured using gas chromatography–mass spectrometry and calculated as nanograms/gram of tissue.

**Ex vivo staining of ROS with fluorescent dyes**. Ex vivo staining was performed as described in our previous papers[21,43]. After euthanasia, 2−3 mm thick kidney slices were immediately rinsed in HBSS, and transferred to DMEM containing 5% FBS, L-Glutamine, pyruvate, sodium bicarbonate, and gentamycin. Ex vivo fresh kidney slices were stained live in the DMEM medium containing 5 μM MitoSOX, 5 μM DCFDA, and 1 μg/ml Hoechst at 37 °C for 30 min. Following washing with DMEM medium, kidney slices were immediately embedded in OCT and frozen. Frozen kidney slices (5 μm) were sectioned within 3 days, and coverslips were placed over sections with VectaShield and imaged with a Leica SP8 confocal microscope.

**Targeted proteomic analysis**. Frozen kidney tissue from 27 mice, including 4 WT, 9 RC/RC, 7 RC/RC + mCAT, 4 RC/null, and 3 RC/null + mCAT, were homogenized in RIPA buffer containing protease inhibitor cocktail. Total proteins were mixed with 200 μL of 1% SDS and 20 μL of BSA internal standard. The mixtures were separated on SDS-PAGE gels, and proteins were extracted from gels. These proteins were determined with a TSQ Quantiva triple quadrupole mass spectrometry system in the selected reaction monitoring mode. The Skyline program was used to determine the integrated area of the appropriate chromatographic peaks for quantification.

**Cell lines and experimental plan**. Immortalized cyst-derived proximal tubular epithelial cells with heterozygous PKD1 mutation (WT 9-7, ATCC® CRL-2830™) and proximal tubular cell line derived from the normal kidney (HK-2, ATCC® CRL-2190™) were cultured with 1:1 DMEM/F12 (Gibco 11320-033) supplemented with 10% FBS, 15 mM HEPES, 50 nM Hydrocortisone (H-6909), 5ug/ml Insulin (I-1882), 5ug/ml Transferrin (T-0665), 50 ng/ml Sodium selenium (S-9133), 1 ng/ml T3 (3,3′5-triiodo-L-thyronine) (T 5516) and 10 ng/ml Epidermal Growth Factor (BD354001). All cells were incubated in an atmosphere of 95% air and 5% CO2 at 37 °C.

**Statistics and reproducibility**. Data were analyzed using Stata IC10 and presented as means ± SEM for normally distributed data, and presented as boxplots (median, 25th and 75th percentiles) for skewed data. To compare differences among groups, a two-sample t-test or ANOVA was used, followed by post hoc tests (Sidak or Tukey method) for significance. $P$ values <0.05 were considered significant.

Penalized linear regression models were fitted to analyze targeted proteomics data by pathway. These models involved protein effects, protein-specific PKD1 effects, protein-specific dose effects, protein-specific mCAT effects, and protein-specific interaction effects of mCAT and dose. The parameters were grouped by proteins, and the composite Minimax Concave Penalty (cMCP) was applied to select important groups and important group members, simultaneously. The reported estimates were derived by fitting unpenalized linear models with the selected effects. Please see supplementary method for detail. In addition, the sample sizes and the number of replicates were described in the text or figure captions.

**Reporting summary**. Further information on research design is available in the Nature Research Reporting Summary linked to this article.

## Data availability
The source data for the targeted proteomics are provided in Supplementary Data 1, https://doi.org/10.6084/m9.figshare.16530663.v2 and raw mass spectrometer files can be accessed through accession number https://doi.org/10.5281/zenodo.5483897. The source data underlying the graphs are provided in Supplementary Data 2. Other relevant data are available from the corresponding author upon request.

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

## Acknowledgements

Funding: NIH K08 HL145138-02 (DFD), AG050886 (VDU, JZ). The proteomics experiments were supported by NIH grants GM137786, GM103447, and AG050911 (MK). We thank the Integrative Redox Biology Core at the Oklahoma Nathan Shock Center (P30 AG050911) for assistance in the measurement of oxidative damage. We thank Drs. Ziying Yan and Zehua Feng for their help in AAV9-mCAT viral production.

## Author contributions

N.D. performed experiments, data analysis, and manuscript writing; A.W.B. performed experiments and data analysis; P.L. performed data analysis (quantitative pathology and immunostaining); F.W. performed data analysis (proteomics data); Y.C. performed experiments; M.T.K. performed experiments (targeted proteomics); G.A.B. and M.S.J. performed experiments (Seahorse assays); V.D. and J.Z. performed Seahorse experimental designs, data analysis, and interpretation and critical revision of the manuscript; K.C. performed data analysis and interpretation (proteomics data); D.D. designed and performed experiments, data analysis, and manuscript writing.

## Competing interests

The authors declare no competing interests.
