## [Transparent Peer Review File · Communications Biology]

Reviewers' comments:

Reviewer #1 (Remarks to the Author):

The work by Daneshgar and co-workers entitled *Metabolic Derangement in Polycystic Kidney Disease Mouse Models Are Ameliorated by Mitochondrial-Targeted Antioxidants* utilizes adeno-associated virus (AAV) technology to ectopically express a mitochondrially-targeted transgene of catalase (mCAT) in two mouse models of ADPKD and examine effects on disease pathology, renal metabolism and oxidative stress, and associated signaling pathways. Catalase is an antioxidant enzyme endogenously expressed in peroxisomes, whose transgenic expression when directed to mitochondria was previously demonstrated to have beneficial effects on lifespan, age-related degeneration, and cardiovascular pathologies in various mouse models. In this manuscript, the authors use both rapidly (RC/null) and slowly (RC/RC) progressive PKD mouse models with the hypomorphic mutant R3277C (RC) allele. They show mCAT expression reduced levels of renal ROS and oxidant damage and ameliorated cystic kidney size and fibrosis, and certain other aspects of cystic disease progression and renal pathology. Targeted proteomics analysis of cystic kidneys showed deficiencies in fatty acid oxidation, TCA cycle, respiratory complexes and endogenous antioxidants, whose levels were partially restored by mCAT expression, although the dramatic reduction of mitochondrial oxygen consumption rate, analyzed in lysates from RC/null kidneys, was not rescued. In addition, treatment of RC/RC mice with a protective mitochondrial peptide, SS31, was able to reduce cystic kidney enlargement and fibrosis in the RC/RC model. Utilizing proximal tubule (PT) epithelial cell culture models of human ADPKD and normal kidneys, they examine the effect of mito-paraquat (superoxide inducer) and mito-Tempo (superoxide scavenger) on cell growth and PKD-related ERK and AMPK signaling effectors in order propose a mechanism for mitochondrially generated ROS as contributing to both cyst proliferation and metabolic derangement in ADPKD pathogenesis.

This manuscript has a number of notable strengths including its attempt to address potential connections between oxidative stress, mitochondrial dysfunction and metabolic alterations in mutant PKD1 cystic kidneys. These represent emerging hallmarks of ADPKD for which we have little understanding and whose illumination could lead to new therapeutic targets and treatments for this disease. The treatment and examination of both rapidly and slowly progressive versions of an orthologous ADPKD mouse model, combined with quantitative measurement of protein (rather than mRNA), of mitochondrial complex activities and of ex vivo oxidant levels are also considered strengths of this work, as is the novel in vivo use of AAV technology as one of the mitochondrially targeted antioxidant treatments.

There are a few issues, however, that have raised questions or concerns (listed below) and should be addressed for further clarification and/or strengthening of the manuscript.

1. In some instance, information was either lacking or not clearly presented:

- (a) What is the age and sexes of RC/null mice utilized for mitochondrial respiratory complex activity? What were the sexes of all mice used for targeted proteomic analyses?
- (b) What is the background of Pkd1-RC and Sox2-Cre mice? What are genotyping methods?
- (c) What is the genotype constituting 'wild type' mice? Do they represent littermates of cystic mice, or were they from a different mouse line, and if so, what is background? For example, Line 130 and Fig 3 refer to RC/WT controls.
- (d) How were cell counts determined in Fig 6a? How many experimental repeats were performed for the cell counts after MPQ or Mtmp treatment? For the Western blots in 6b?
- (e) How were the blot quantification graphs in Fig 6 b, c, and d obtained (y-axes lack labels)? Specifically, were phospho-AMPK and phospho-ERK levels normalized to total AMPK and total ERK levels, respectively? Was GAPDH used in normalization, and if so, how? How many cell lysates and kidney lysates were analyzed? Are blots shown in b, c, and d representative? The GAPDH blot in 6d is not of good quality.
- (f) Please make clear the numbers of samples analyzed for each of the figure legends by including the

"n" in each subsection of the legend (e.g., Fig 8).

(g) Scale bars should be added to all photomicrographs.

2. (a) What was the effect of AAV9 vector and AAV9-mCAT treatments on the body weight (alone) of the RC/RC and RC/null mice (versus untreated)? This information is important for assessing potential adverse effects of treatment.

(b) What does the delivery of mCAT into RC/null kidneys look like (i.e., CAT and VDAC staining as in Fig 1g)? In particular, is mCAT expression evident in cystic epithelium?

3. (a) Decreased mitochondrial complex activity was demonstrated for the rapidly progressive RC/null kidneys only, and presumably from a late stage timepoint (comment 1a above). In order to invoke mitochondrial dysfunction as an important driver of disease progression (rather than of disease severity), these analyses would need to be performed at different ages/stages of disease with the RC/null kidneys, and better yet, with RC/RC kidneys.

(b) When, in the course of disease progression, do RC/RC kidneys exhibit mitochondrial dysfunction?

(c) Could the reduction in mitochondrial mass in RC/null kidneys be a result of PT dilution by cystic distal/collecting tubules?

4. The authors may want to consider reorganization of the Results section and combining data into fewer figures. This could help clarity, remove redundancies within the text and allow shortening of the manuscript. For example, it appears that the data shown in Fig 1d for RC/null kidney lysates is part of/the same as that shown in Fig 5c and d. The effect of SS31 treatment on oxidant levels and damage (Fig 7) is presented before the treatment is described in the text for Fig 8. Examples of redundant sentences/information: Lines 52-56 in Supplementary Methods; Lines 170 - 175 are repetitive with introduction section. Perhaps grouping RC/RC data together in one figure with RC/null data in a separate figure might be considered.

5. The results provided in Figure 7 are critically important for supporting the mitochondria as a major source of reactive oxidants within cystic kidneys, and additional details are needed to ascertain their validity and to enable repetition by other investigators.

(a) Please describe in detail the methodology utilized to ensure equivalent comparisons between treatments for ex vivo quantification. For example, the Johnson and Rabinovitch reference (ref 34) describes multiple approaches for quantitative analysis.

(b) How many microscopic fields (or tubules or cells) were quantified for the graph data presented?

(c) How many kidneys of each condition/treatment were imaged and quantified for the graphs presented?

(d) Images in column b of Fig 7 appear to be proximal tubules, which raises the questions of where within the kidney sections and what tubule segments are measurements of mitochondrial and total cellular ROS being quantified?

(e) Do cystic epithelium exhibit more ROS than non-cystic tubules within RC/RC kidneys?

(f) Did mCAT expression reduce oxidant levels and oxidant damage in RC/null kidneys?

6.(a) Provide the rationale for performing mechanistic analyses in proximal tubule cell lines when the RC mouse model is characterized by cysts of distal/CD tubules.

(b) Does treatment of the WT9-7 cell line with SS31 or mCAT 'rescue' cell counts to HK2 cell levels?

(c) Were phosphoAMPK and phosphoERK levels elevated in WT9-7 cells + saline compared to HK2 control cells + saline (i.e., in the basal state), and/or do the PKD cells respond differently than HK2 cells to Mtmp and MPQ treatments?

(d) The authors should exercise caution when concluding that Mtmp or MPQ treatment altered the 'proliferation rate' of PKD1 mutant cells (i.e., lines 229-231) as cell count does not demonstrate a change in proliferation rate but could just as well result from increases or decreases in survival cell survival. Conclusions regarding effects on cell proliferation or proliferation rate would require additional experiments (e.g., proliferation markers, cell cycle analysis, or measurement of cell death, etc).

7. (a) Did either mCAT or SS31 treatment reduce standard markers of cell proliferation (e.g., PCNA by IH or IF) in RC/RC kidneys?
(b) Did either treatment alter phosphoAMPK levels or mTOR activity markers (e.g., phospho-mTOR, -S6kinase) in kidney lysates or in cystic epithelium?
(c) Were the effects of mito-targeted treatments and resulting disease amelioration specific to cystic tubules, or perhaps to non-cystic proximal tubular cells? For example, do mCAT and SS31 differentially target specific tubule segments, e.g., proximal tubules which have more mitochondrial density? Such information could greatly impact our view of the pathogenetic mechanism of PKD and of the mechanism of a mitochondrially targeted antioxidant therapeutic.

8. (a) The reference provided as previous support of the critical role of ROS-induced mitochondrial damage in cystic disease caused by PKD1 mutation (line 243; Ref 15) is incorrect. The PKD1 protein in Ref 15 refers to protein kinase D.

(b) The concluding sentences: "The mechanisms are mediated by better preservation of AMPK and the fatty acid oxidation pathway, and mitigation of ROS-mediated ERK1/2 (Extracellular signal-regulated protein kinases 1 and 2) phosphorylation" (lines 81-83); "These findings indicate the role of mitochondrial ROS in stimulating cell proliferation via ERK phosphorylation, and the inhibitory effect of mitochondrial ROS on AMPK, one of the master regulators of cellular metabolism" (lines 237-239); and "The mechanisms include mitigation of ERK1/2 phosphorylation, preservation of AMPK and partial reversal of metabolic derangement in PKD1" (lines 354-355), are overselling the results of these studies by emphasizing circumstantial connections that were not demonstrated. For example, a "preservation of AMPK" was not demonstrated in kidneys of mice treated with either mCAT or SS31, and a direct link between AMPK and FAO or ROS-induced pERK and cyst-lining cell proliferation was not demonstrated. It would be more appropriate to temper these statements (e.g., These findings suggest/are consistent with a role...; A potential mechanism involving...).

9. The discussion section could include more relation to previous studies. For example, how does the increase in survival time with mCAT in RC/null mice compare with previous mCAT studies in other diseases? Does this represent a general (non-PKD specific) effect of mitochondrially-targeted CAT overexpression on lifespan and/or cardiac pathology as reported previously by the authors? How do these results relate to other studies demonstrating metabolic alterations in ADPKD, e.g., Warburg effect/aerobic glycolysis? How do these results relate to other studies in PKD using nontargeted antioxidant treatments, and in light of recent observations (e.g., Lu Y et al. *Sci Transl Med* 2020 Jul 29;12(554):eaba3613).

10. Errors noted during review include:

- a) The verb "Are" in the manuscript title does not agree with the singular topic (Derangement) and should be the singular form, i.e., "Is".
- b) The labels on x-axis (mitochondrial complex) are different between Figures 1d, 1e and 5c.
- c) In Figure 3 legend, lines 612-613, phrases and information referring to high doses should be removed from this legend and added to Supp Fig 1 legend. Lines 3-4 of Supp Figure 1 legend should be added to Fig 3 legend, and significant relationships added to Supp Figure 1.
- d) In legend for Fig 7, line 293 should include both Fig 7E and F.
- e) Lines 193 and 194, PKD1 is misspelled (PDK1).

Reviewer #2 (Remarks to the Author):

In this study Daneshgar and colleagues report that a mitochondrial targeted antioxidant is able to improve disease progression in PKD.

The study has some interesting points, especially from a translational point of view. However, there is

very limited novelty, there are some technical flaws (see below) and more importantly, it is unclear why the authors should ignore completely the extensive previous work published on the topic.

Specific comments:

Figure 1-E the authors show a reduced mitochondrial activity and complexes assembly in Pkd mutant kidneys. It is unclear why the authors completely ignore previous studies showing similar analysis (Cassina et al, 2020). How do the results compare with the null kidneys shown in the previous study?

In figure 2 the authors show an amelioration of the phenotype when using the AAV-mCAT vectors in the RC/RC mouse model: authors should show the data with individual points for each animal analyzed.

Is figure 2a showing a representative example or an extreme case?

Figure 2 d lacks indication of the Yaxis. Is BUN shown as mg/dl? Usually it has to be above 50 to be considered as a good indicator of loss of renal function. So, it is unclear what it means a value that goes from 17 to 11.

Figure 3a is it representative or extreme case? Again, the survival curves in 3d suggest that numerous animals were analyzed, so it would be appropriate to show the individual values in figures 2b and c.

Figure 5: TCA cycle was previously involved in PKD (Podrini et al, Comms Biol, 2018). Beta oxidation reduction was shown by two groups (Menezes et al, eBioMedicine, 2016;; Hajarnis et al, Nat Comm, 2017). And similarly mitochondrial genes.

Indeed, involvement of mitochondria in PKD (including ROS: Lu et al, Science Translational Medicine, 2020; Irazabal and Torres, Int Journal of Molecular Sciences, 2020) has been extensively shown and the authors do not cite any of the studies. Here are some:

DOI:

10.3390/ijms21061994

10.1038/s42003-018-0200-x

10.1096/fj.201901739RR

10.1091/mbc.E16-08-0597

10.1038/s41598-018-20856-6

10.1126/scitranslmed.aba3613

Reviewer #3 (Remarks to the Author):

The manuscript by Daneshgar et al. documents mitochondria dysfunction in ADPKD and mitochondria protective agents as potential therapeutics for ADPKD. The authors discovered a decreased mitochondrial complexes activity and downregulation of mitochondrial and metabolic proteins involved in the TCA cycle, fatty acid oxidation (FAO), respiratory complexes. Functional analysis indicated that mitochondrial-targeted catalase (mCAT) overexpression or mitochondrial protective peptide (SS31) treatment slows cystogenesis in ADPKD mouse models. Overall, this is a potentially interesting study. However, the following issue need to be solved before the publication of this manuscript can be considered.

The mitochondria defects in ADPKD are well documented in previous publications. Similar therapy (MitoQ) targeting mitochondria in human ADPKD cell model was described as in Ishimoto et al., Mol. Cell Bio., 2017 [PMID: 28993480]. The alteration of mitochondrial proteomics was also recently reported by Lu et al., Sci Transl Med., 2020 [PMID: 32727915]. Therefore, the major new finding in this study is the in vivo application of mitochondrion-targeted therapy for ADPKD. The authors show

that intraperitoneal injection of AAV9-mCAT can alleviate the progression of ADPKD. However, gene delivery to kidney cells from the blood is inefficient because of the natural filtering functions of the glomerulus. The authors showed that the catalase expression was increased in kidney tubular epithelial cells with low dose injection of AAV9-mCAT (Fig. 1G). As shown in their proteomics analysis, catalase expression decreased in PKD1 mutant mice comparing to WT mice (Fig. 5A). Thus, the upregulation of catalase may be due to disease remission rather than delivery of AAV9-mCAT to kidney cystic epithelial cells. Adding a tag to AAV9-mCAT construct and examining with anti-tag antibody would be a straightforward way to rule out the above possibility and to identify the primary target organs of AAV9-mCAT. Co-staining with segment-specific markers would also be helpful for determining the targeting cells of mCAT. Furthermore, the authors performed immunostaining in 6-month-old RC/RC mice, in which cysts should be easily observed. However, these were not shown in their immunostaining data (Fig. 1G and 7A).

Reviewer #1 (Remarks to the Author):

The work by Daneshgar and co-workers entitled *Metabolic Derangement in Polycystic Kidney Disease Mouse Models Are Ameliorated by Mitochondrial-Targeted Antioxidants* utilizes adeno-associated virus (AAV) technology to ectopically express a mitochondrially-targeted transgene of catalase (mCAT) in two mouse models of ADPKD and examine effects on disease pathology, renal metabolism and oxidative stress, and associated signaling pathways. Catalase is an antioxidant enzyme endogenously expressed in peroxisomes, whose transgenic expression when directed to mitochondria was previously demonstrated to have beneficial effects on lifespan, age-related degeneration, and cardiovascular pathologies in various mouse models. In this manuscript, the authors use both rapidly (RC/null) and slowly (RC/RC) progressive PKD mouse models with the hypomorphic mutant R3277C (RC) allele. They show mCAT expression reduced levels of renal ROS and oxidant damage and ameliorated cystic kidney size and fibrosis, and certain other aspects of cystic disease progression and renal pathology. Targeted proteomics analysis of cystic kidneys showed deficiencies in fatty acid oxidation, TCA cycle, respiratory complexes and endogenous antioxidants, whose levels were partially restored by mCAT expression, although the dramatic reduction of mitochondrial oxygen consumption rate, analyzed in lysates from RC/null kidneys, was not rescued. In addition, treatment of RC/RC mice with a protective mitochondrial peptide, SS31, was able to reduce cystic kidney enlargement and fibrosis in the RC/RC model. Utilizing proximal tubule (PT) epithelial cell culture models of human ADPKD and normal kidneys, they examine the effect of mito-paraquat (superoxide inducer) and mito-Tempo (superoxide scavenger) on cell growth and PKD-related ERK and AMPK signaling effectors in order propose a mechanism for mitochondrially generated ROS as contributing to both cyst proliferation and metabolic derangement in ADPKD pathogenesis.

This manuscript has a number of notable strengths including its attempt to address potential connections between oxidative stress, mitochondrial dysfunction and metabolic alterations in mutant PKD1 cystic kidneys. These represent emerging hallmarks of ADPKD for which we have little understanding and whose illumination could lead to new therapeutic targets and treatments for this disease. The treatment and examination of both rapidly and slowly progressive versions of an orthologous ADPKD mouse model, combined with quantitative measurement of protein (rather than mRNA), of mitochondrial complex activities and of ex vivo oxidant levels are also considered strengths of this work, as is the novel in vivo use of AAV technology as one of the mitochondrially targeted antioxidant treatments.

There are a few issues, however, that have raised questions or concerns (listed below) and should be addressed for further clarification and/or strengthening of the manuscript.

1. In some instance, information was either lacking or not clearly presented:

(a) What is the age and sexes of RC/null mice utilized for mitochondrial respiratory complex activity? What were the sexes of all mice used for targeted proteomic analyses?

Reply: Information for mice used for mitochondrial respiratory complex activity and targeted proteomic added to figure legend 1 and 5. We used 11 WT (5 female, 6 male), 3 RC/+ (1F, 2M), 4 RC/null mice and 4 RC/null treated with mCAT (2F and 2M per group), 6 RC/RC (6M) and 6 RC/RC treated with mCAT (1F, 5M). There was no gender difference in any of the phenotypes, so the data from both genders were used. RC/null mice were aged 21 days and RC/RC mice were 6-8 months for mitochondrial complex activity. Also, mice used for targeted proteomic analysis were as followed: 3 female and 6 male RC/RC, 5 male and 2 female RC/RC treated with AAV9-mCAT, 2 male and 2 female RC/null, 2 male and 1 female RC/null treated with AAV9-mCAT and 2 female and 2 WT male.

(b) What is the background of Pkd1-RC and Sox2-Cre mice? What are genotyping methods?

Reply: Both Sox2-Cre and Pkd1-RC mice are from C57BL/6 genetic background. Sox2-Cre ($Edi13^{Tg(Sox2-cre)1Amc}$) were genotyped using standard PCR assay for Edil3 transgenic mice following the Jackson laboratory genotyping protocol 31762. Pkd1-RC genotyping were conducted using these two primers:

5'-CAA AGG TCT GGG TGA TAA CTG GTG-3'

5'-CAG GAC AGC CAA ATA GAC AGG G-3'

(c) What is the genotype constituting 'wild type' mice? Do they represent littermates of cystic mice, or were they from a different mouse line, and if so, what is background? For example, Line 130 and Fig 3 refer to RC/WT controls.

Reply: We have used two different controls. Wild type (WT) from a different mouse line of the same C57Bl6/J background as controls for RC/RC mice. $PKD1^{RC/WT}$ were the littermates of $PKD1^{RC/null}$. Complete information about these mice were added in lines 384-388. "The $PKD1^{RC/null}$ and RC/WT control mice were generated by crossing $PKD1^{RC/RC}$ mice with $PKD1^{+/-}$ mice. The $PKD1^{+/-}$ were generated by breeding $PKD1^{flox/flox}$ mice (B6.129S4-*Pkd1*^{tm2Ggg}/J, JAX, 129S4/SvJae background) with germline Sox2-Cre transgenic mice ($Edi13^{Tg(Sox2-cre)1Amc}$, JAX, C57BL/6 background)".

Of note, there was no observed difference in any of the phenotypes between WT and RC/WT.

(d) How were cell counts determined in Fig 6a? How many experimental repeats were performed for the cell counts after MPQ or Mtmp treatment? For the Western blots in 6b?

Reply: Cell counts were performed using Fiji imageJ plugin, automated cell counting of single-color images. We performed independent experiments three times for each of treatment and control groups for both cell counting and western-blot. Information for these were added to Figure 6 legend.

(e) How were the blot quantification graphs in Fig 6 b, c, and d obtained (y-axes lack labels)? Specifically, were phospho-AMPK and phospho-ERK levels normalized to total AMPK and total ERK levels, respectively? Was GAPDH used in normalization, and if so, how? How many cell lysates and kidney lysates were analyzed? Are blots shown in b, c, and d representative? The GAPDH blot in 6d is not of good quality.

Reply: We have done more than 10 additional immunoblots to obtain higher quality images. Each western blot has been repeated at least three times. All the blots that were shown were the closest to the average, i.e. "representative blots". The Y-axes labels were now included in the graphs. The phospho-AMPK and phospho-ERK levels were normalized to total AMPK and total ERK, respectively. GAPDH was only used as loading control in Figure 6 and the GAPDH showed roughly equal loading for all blots. Three separate experiments were conducted for cell lysates and three kidney lysates for control group and 6-9 for treatment groups were used for analysis. GAPDH blot in the original Fig 6d (now as Fig 6g) has been replaced with a better-quality blot.

(f) Please make clear the numbers of samples analyzed for each of the figure legends by including the "n" in each subsection of the legend (e.g., Fig 8).

Reply: Number of analyzed samples (n) added for all figures.

(g) Scale bars should be added to all photomicrographs.

Reply: Scale bars are now added to all photomicrographs.

2. (a) What was the effect of AAV9 vector and AAV9-mCAT treatments on the body weight (alone) of the RC/RC and RC/null mice (versus untreated)? This information is important for assessing potential adverse effects of treatment.

Reply: A graph illustrating the body weight of WT, RC/RC and RC/null mice untreated or treated with AAV9 vector or AAV9-mCAT were added as Supplementary Figure 4c. We did not observe any significant difference in bodyweight of different treatment groups.

(b) What does the delivery of mCAT into RC/null kidneys look like (i.e., CAT and VDAC staining as in Fig 1g)? In particular, is mCAT expression evident in cystic epithelium?

Reply: We additionally performed immunofluorescence staining for RC/null mice treated with mCAT (Figure 2b). Mitochondrial targeted catalase is more prominent in intact tubules and occasionally seen in cystic epithelium (supplementary Figure 3d-e).

3. (a) Decreased mitochondrial complex activity was demonstrated for the rapidly progressive RC/null kidneys only, and presumably from a late stage timepoint (comment 1a above). In order to invoke mitochondrial dysfunction as an important driver of disease progression (rather than of disease severity), these analyses would need to be performed at different ages/stages of disease with the RC/null kidneys, and better yet, with RC/RC kidneys.

Reply: This is a critical point. As suggested, we have additionally measured mitochondrial complexes activity in RC/RC mice. As presented in Figure 1, citrate synthase activity was decreased by more than 50% in lysates from RC/null and ~30% in lysates from RC/RC kidneys compared to controls (Fig.1b, f), suggesting a decrease in mitochondrial mass. There were trends of decreased complex III and IV ($p=0.09$ and $p=0.07$ respectively, Fig.1h) in RC/RC kidneys, which were not significant after normalization to citrate synthase (Figure 1i), suggesting that the decreased mitochondrial mass was the main mechanisms of mitochondrial impairment. However, given the variability, the sample size of 6, and $\alpha=0.05$, the powers were 24% for complex I and ~50% for complexes II-IV, which were underpowered to detect the observed difference in RC/RC vs WT kidneys). All results have been added to Figure 1 (f-i) and interpretation added in the appropriate section (lines 98-100, 116-120)

(b) When, in the course of disease progression, do RC/RC kidneys exhibit mitochondrial dysfunction?

Reply: As discussed above, we observed ~30% decrease in citrate synthase activity in kidneys from 6-8-month-old RC/RC mice with slowly progressive disease and >50% decrease in citrate synthase in kidneys from 21-day-old RC/null with rapidly progressive disease. Most respiratory complexes activity was also decreased (but not significant) in RC/RC. It is possible that RC/RC kidneys may develop more prominent mitochondrial dysfunction at the later stage of RC/RC (e.g. at 12 month-of-age, which was not studied in the current project given the high cost associated with long-term maintenance of mouse colony). However, we demonstrated that the

RC/null had dramatic mitochondrial dysfunction, including reduced mitochondrial mass and decreased activity of respiratory complexes.

(c) Could the reduction in mitochondrial mass in RC/null kidneys be a result of PT dilution by cystic distal/collecting tubules?

Reply: We agree that this is possible and have revised the manuscript as following:

“Furthermore, although the positive correlation between loss of mitochondrial mass and the progression of cysts in two models of PKD1 does not provide evidence of causality, we propose that there is a “vicious cycle” between mitochondrial dysfunction, increased ROS leading to the progression of cysts, which may subsequently induce further decrease in mitochondrial function. The replacement of tubules by cysts and fibrosis may also contribute to further reduction of mitochondrial mass and progressive decline in mitochondrial function. The fact that scavenging mitochondrial ROS by mCAT ameliorated mitochondrial and metabolic proteome, attenuated cyst progression and prevented the decrease in mitochondrial mass in RC/RC kidneys (borderline significance) suggest that breaking the vicious cycle may delay PKD1 progression.”

4. The authors may want to consider reorganization of the Results section and combining data into fewer figures. This could help clarity, remove redundancies within the text and allow shortening of the manuscript. For example, it appears that the data shown in Fig 1d for RC/null kidney lysates is part of/the same as that shown in Fig 5c and d. The effect of SS31 treatment on oxidant levels and damage (Fig 7) is presented before the treatment is described in the text for Fig 8. Examples of redundant sentences/information: Lines 52-56 in Supplementary Methods; Lines 170 – 175 are repetitive with introduction section. Perhaps grouping RC/RC data together in one figure with RC/null data in a separate figure might be considered.

Reply: As suggested, we grouped RC/RC data together in one figure (Figure 3) and RC/null data in a separate figure (Figure 4) and eliminated the previous Figure 4. Fig 5c and d were moved to supplementary figures as SFig. 5a-b. Figure 7 and 8 interchanged so that the description of SS31 treatment would be before the presentation of its effect on oxidant levels and damage.

5. The results provided in Figure 7 are critically important for supporting the mitochondria as a major source of reactive oxidants within cystic kidneys, and additional details are needed to ascertain their validity and to enable repetition by other investigators.

(a) Please describe in detail the methodology utilized to ensure equivalent comparisons

between treatments for ex vivo quantification. For example, the Johnson and Rabinovitch reference (ref 34) describes multiple approaches for quantitative analysis.

Reply: As described in the Johnson and Rabinovitch reference, since this is an ex vivo live staining, the time from euthanasia to live staining is critical. It was ensured that each kidney section was sectioned the same way, stained at the same time post-euthanasia and stained for the same duration and scoped within two days. Each experimental group include WT, RC/RC and RC/RC mCAT or RC/RC+SS31. All tissues were sectioned and scoped the same day for each experimental group, to avoid batch effect. No comparison has been done for experiments done from different days.

(b) How many microscopic fields (or tubules or cells) were quantified for the graph data presented?

Reply: At least 10-15 microscopic fields were quantified for each group.

(c) How many kidneys of each condition/treatment were imaged and quantified for the graphs presented?

Reply: Three to four mice from each experimental group were imaged and quantified.

(d) Images in column b of Fig 7 appear to be proximal tubules, which raises the questions of where within the kidney sections and what tubule segments are measurements of mitochondrial and total cellular ROS being quantified?

Reply: We agree with the reviewer that the tubules in Fig.7b are likely proximal tubules. Staining with MitoSox and DCFDA cannot easily distinguish proximal vs distal tubules (particularly when the signals were weak). Given the time-sensitive nature of live staining (unfixed) and the incompatibility with other staining, it is not feasible to add additional staining that may require additional processing, such as fixation. As an alternative way to answer this question, we performed IHC staining for nitrotyrosine (NT), a marker for peroxynitrite formation (oxidative damage), and quantified the intensity of proximal vs. distal tubules. As shown in the new Figure 8g-k, there is no significant difference between nitrotyrosine staining in the proximal and distal tubules in RC/RC, however, the NT is significantly stronger in pericycystic epithelial lining. Also, treatment with mCAT reduced NT in both proximal and distal tubules of RC/RC mice, but this reduction was only significant in proximal tubules, consistent with the fact that proximal tubules have more mitochondria than distal tubules. NT staining was slightly attenuated in cystic epithelial lining in RC/RC kidneys treated with mCAT ($p=0.08$) (Fig.

8j-k). Since RC/null kidneys were mostly consisted of cysts and fibrotic tissue, comparison between RC/null and RC/null treated with mCAT was not feasible (SFig. 8). This has been added to the result section.

(e) Do cystic epithelium exhibit more ROS than non-cystic tubules within RC/RC kidneys?

Reply: As evident by IHC staining of nitrotyrosine, cystic lining epithelium exhibit increase staining of nitrotyrosine compare with non-cystic tubules and treatment with mCAT slightly attenuated NT staining ($p=0.08$) (Fig. 8g-k)

(f) Did mCAT expression reduce oxidant levels and oxidant damage in RC/null kidneys?

Reply: Yes, we have performed additional F2 isoprostane measurement to show that treatment with AAV9-mCAT significantly reduced F2 isoprostane in these mice (shown in fig 7f). Additionally, the NT staining discussed in d-e is consistent with the isoprostane data.

6.(a) Provide the rationale for performing mechanistic analyses in proximal tubule cell lines when the RC mouse model is characterized by cysts of distal/CD tubules.

Reply: Although cysts may originate from distal/CD tubules, it also involves proximal tubules. We chose proximal tubule cell line as they have more abundant mitochondria, consistent with our focus of the study. Furthermore, it has been used in previous report (Ishimoto et al, MCB 2017).

(b) Does treatment of the WT9-7 cell line with SS31 or mCAT 'rescue' cell counts to HK2 cell levels?

Reply: Yes, treatment with mCAT or SS31 decreased WT9-7 cell count to the levels comparable to that in HK2 cell levels ($p=0.002$ for mCAT and 0.155 for SS31. Representative images of the newly performed experiments in WT9-7 cells treated with saline, mCAT or SS31 are now presented in Figure 6a.

(c) Were phosphoAMPK and phosphoERK levels elevated in WT9-7 cells + saline compared to HK2 control cells + saline (i.e., in the basal state), and/or do the PKD cells respond differently than HK2 cells to Mtmp and MPQ treatments?

Reply: We additionally performed cell experiments using both WT9-7 and HK2 cells, and treated HK2 with saline, Mtmp or MPQ. At basal state (saline treated group), p-ERK was higher, and p-

AMPK was lower in WT9-7 cells than in HK2 control cells (sfig.7a). The response of HK2 cells to MPQ and Mtmp is similar but in less magnitude compared with the response seen in WT9-7 cells. Treatment with MPQ increased p-ERK and decreased p-AMPK in HK-2 cells. Conversely, treatment with Mtmp decreased the p-ERK and increased p-AMPK with larger variability (not significant). Overall, the response of PKD1 cells to inducer or scavenger of mitochondrial superoxide is stronger than that seen in control cells.

(d) The authors should exercise caution when concluding that Mtmp or MPQ treatment altered the 'proliferation rate' of PKD1 mutant cells (i.e., lines 229-231) as cell count does not demonstrate a change in proliferation rate but could just as well result from increases or decreases in survival cell survival. Conclusions regarding effects on cell proliferation or proliferation rate would require additional experiments (e.g., proliferation markers, cell cycle analysis, or measurement of cell death, etc).

Reply: We agree with the reviewer and have additionally performed Ki-67 immunostaining as a proliferative marker. The data is now presented in Fig 6b (see below), which demonstrate a substantial increase in Ki-67 positive nuclei after MPQ treatment and a decrease in Ki-67 positive nuclei after Mtmp treatment in PKD1. In addition, there was no obvious changes in cell morphology or cell death in any of the experimental groups.

7. (a) Did either mCAT or SS31 treatment reduce standard markers of cell proliferation (e.g., PCNA by IH or IF) in RC/RC kidneys?

Reply: We performed IHC staining for Ki67, one of the markers of cell proliferation, and showed that ki-67 staining is increased in RC/RC and RC/null kidney tissues compared with WT and is decreased when treated with mCAT, but none of these changes were significant (SFig. 6), likely due to variability and surprisingly low numbers of Ki-67 positive nuclei *in vivo*. Lower Ki-67 staining in RC/null tubules compared with RC/RC tubules can be explained by the fact that most of RC/null tissue at this stage has been replaced with either cysts or fibrotic tissue.

(b) Did either treatment alter phosphoAMPK levels or mTOR activity markers (e.g., phospho-mTOR, -S6kinase) in kidney lysates or in cystic epithelium?

Reply: We performed immunoblotting for p-AMPK in kidney lysates of RC/RC, RC/null mice and RC/RC, RC/null treated with AAV9-mCAT and showed that treatment with mCAT increase p-AMPK levels in both mouse models of PKD (SFig. 7b). Due to low abundance and instability of p-AMPK in kidney tissues, we were only able to observe the bands in a freshly prepared protein lysates (using 4x phosphatase inhibitors) from fresh kidney tissues. We have tried ~8 times to

probe additional frozen tissue or frozen proteins but failed to detect the phospho-epitope of AMPK (using the most used antibody from Cell Signaling Technology, Thr172, #2531). Since repeating animal experiments will take at least 6 months, we are not able to increase our sample size at this point.

(c) Were the effects of mito-targeted treatments and resulting disease amelioration specific to cystic tubules, or perhaps to non-cystic proximal tubular cells?

Reply: We have additionally performed IHC for nitrotyrosine. The new data is presented in Fig 8g-k and described in the result section, as following:

“To investigate the extent of tissue oxidative damage, we performed IHC staining for nitrotyrosine (NT), a marker of peroxynitrite formation, and analyzed NT staining intensity in proximal and distal tubules as well as epithelial cells lining the enlarged cysts. There was ~70% increase in NT staining intensity in both proximal and distal tubules of RC/RC kidneys compared with WT controls ($p < 0.001$ for both). While there was no significant difference in NT staining between proximal and distal tubules ($p = 0.369$), there was significantly higher NT staining in cystic epithelial cells than in proximal or distal tubules of RC/RC kidneys ($p < 0.001$, Fig. 8h-k), indicating higher oxidative damages in cystic tubules. Treatment with AAV9-mCAT decreased NT staining in both proximal and distal tubules of RC/RC kidneys, although this was only significant in the proximal tubules ($p < 0.001$) but not distal tubules ($p = 0.12$), consistent with the fact that proximal tubules have more mitochondria. NT staining was slightly attenuated in cystic epithelial lining in RC/RC kidneys treated with mCAT ($p = 0.08$) (Fig. 8j-k). Since RC/null kidneys were mostly consisted of cysts and fibrotic tissue, comparison between RC/null and RC/null treated with mCAT was not feasible (SFig. 8).”

For example, do mCAT and SS31 differentially target specific tubule segments, e.g., proximal tubules which have more mitochondrial density? Such information could greatly impact our view of the pathogenetic mechanism of PKD and of the mechanism of a mitochondrially targeted antioxidant therapeutic.

Reply: To answer this question, we further performed co-immunostaining of frozen kidney sections of both RC/RC, RC/null mice and RC/RC, RC/null mice treated with AAV9-mCAT with specific proximal or distal tubular markers (LTL and DBA, respectively) and VDAC and catalase as shown in supplementary figure 1a-e. Our results showed that there is no obvious difference between proximal and distal tubule in terms of mCAT delivery. The stronger protection in proximal than distal tubules by mCAT (as shown above) is consistent with the fact that proximal tubules have more mitochondria.

8. (a) The reference provided as previous support of the critical role of ROS-induced mitochondrial damage in cystic disease caused by PKD1 mutation (line 243; Ref 15) is incorrect. The PKD1 protein in Ref 15 refers to protein kinase D.

Reply: Thanks for bringing up this issue. That reference was removed.

(b) The concluding sentences: “The mechanisms are mediated by better preservation of AMPK and the fatty acid oxidation pathway, and mitigation of ROS-mediated ERK1/2 (Extracellular signal-regulated protein kinases 1 and 2) phosphorylation” (lines 81-83); “These findings indicate the role of mitochondrial ROS in stimulating cell proliferation via ERK phosphorylation, and the inhibitory effect of mitochondrial ROS on AMPK, one of the master regulators of cellular metabolism” (lines 237-239); and “The mechanisms include mitigation of ERK1/2 phosphorylation, preservation of AMPK and partial reversal of metabolic derangement in PKD1” (lines 354-355), are overselling the results of these studies by emphasizing circumstantial connections that were not demonstrated. For example, a “preservation of AMPK” was not demonstrated in kidneys of mice treated with either mCAT or SS31, and a direct link between AMPK and FAO or ROS-induced pERK and cyst-lining cell proliferation was not demonstrated. It would be more appropriate to temper these statements (e.g., These findings suggest/are consistent with a role...; A potential mechanism involving...).

Reply: These concluding sentences were revised as following:

(lines 83-85) The potential mechanisms are mediated by increase in AMPK and the fatty acid oxidation pathway, and mitigation of ROS-mediated ERK1/2 (Extracellular signal-regulated protein kinases 1 and 2) phosphorylation.

(lines 289-292) These findings suggest the role of mitochondrial ROS in stimulating cell proliferation via ERK phosphorylation, and the inhibitory effect of mitochondrial ROS on AMPK, one of the master regulators of cellular metabolism.

(lines 455-458) The potential mechanisms include mitigation of ERK1/2 phosphorylation, preservation of AMPK and partial reversal of metabolic derangement in PKD1

9. The discussion section could include more relation to previous studies. For example, how does the increase in survival time with mCAT in RC/null mice compare with previous mCAT studies in other diseases? Does this represent a general (non-PKD specific) effect of mitochondrially-targeted CAT overexpression on lifespan and/or cardiac pathology as reported previously by the authors?

Reply: We have added these in the discussion:

“Schriner et al generated mice overexpressing catalase targeted to mitochondria (mCAT) or the natural sites within peroxisomes (pCAT). The beneficial effect of mCAT exceeded the effect of pCAT overexpression in term of murine lifespan extension, cardiac hypertrophy and failure.”

Previous mCAT studies have focused on amelioration of pathology or disease progression in various age-related diseases and cancers. In this study, since RC/null mice developed uremia and died between post-natal 18-25 days, the lifespan extension effect of mCAT provided a strong evidence of survival benefit by delaying the onset of kidney failure in this rapidly progressive PKD model. The lifespan extension reported in Science 2005 was for multiple aging cohorts (up to 36 months), while the PKD1 RC/null mice only survived for 18-30 days and died of uremia (BUN>100). The mCAT effect emphasize the benefit of targeting (protecting) mitochondria from oxidative stress, which has been implicated in several diseases, including PKD1.

How do these results relate to other studies demonstrating metabolic alterations in ADPKD, e.g., Warburg effect/aerobic glycolysis?

Reply: We have added a brief discussion comparing our proteomics data with the comprehensive metabolomics study by Podrini et al, Communication Biology 2018, as following (lines 393-402):

“In this study, we applied a state-of-the-art targeted proteomics approach to measure the abundance of 137 proteins involved in several mitochondrial and metabolic pathways as well as those involved in the peroxisomal and proteostasis pathways. We demonstrated that the top three pathways significantly suppressed by the effect of PKD1 mutations were the TCA cycle (-31%), FAO (-29%), and respiratory complexes (-28%). The latter was confirmed by substantial decreases in mitochondrial respiratory complex activity (Fig.1a-). Interestingly, the pattern of proteome changes in the current study is reinforced by a comprehensive metabolomics study, which reported that the most striking changes of metabolites in PKD1 suggested a profound impairment of TCA cycle, FAO and fatty acid synthesis, glycolysis as well as pentose phosphate pathway³¹.”

How do these results relate to other studies in PKD using nontargeted antioxidant treatments, and in light of recent observations (e.g., Lu Y et al. Sci Transl Med 2020 Jul 29;12(554):eaba3613).

Reply: We have cited this article, as following:

Lines 355-359 “A recent study using proteomics and transcriptomics approach reported that NRF2-regulated antioxidant pathways were suppressed in ADPKD, while the activation of NRF2 ameliorated oxidative stress and cystogenesis in ADPKD25. This suggests that oxidative stress play a crucial role in cyst formation/progression 23,26,27,28.”

NRF2 is a master regulator of endogenous antioxidants, including non-mitochondrial antioxidants and glutathione reductase/peroxidase system, which also detoxify hydrogen peroxide within mitochondria. Indeed, this paper reinforce our findings, but we focus on mitochondrial ROS, and they focus on endogenous antioxidant pathways.

10. Errors noted during review include:

a) The verb “Are” in the manuscript title does not agree with the singular topic (Derangement) and should be the singular form, i.e., “Is”.

Reply: This issue has been corrected. The new title read as “Metabolic Derangement in PKD1 Mouse Models Is Ameliorated by Mitochondrial-Targeted Antioxidants”.

b) The labels on x-axis (mitochondrial complex) are different between Figures 1d, 1e and 5c.

Reply: All figures 1d, 1e and 5c relabeled to exact same labels.

c) In Figure 3 legend, lines 612-613, phrases and information referring to high doses should be removed from this legend and added to Sup Fig 1 legend. Lines 3-4 of Supp Figure 1 legend should be added to Fig 3 legend, and significant relationships added to Supp Figure 1.

Reply: Figure 3 and Sup Fig 1 legend relabeled.

d) In legend for Fig 7, line 293 should include both Fig 7E and F.

Reply: This issue has now been corrected.

e) Lines 193 and 194, PKD1 is misspelled (PDK1).

Reply: This issue has now been corrected.

Reviewer #2 (Remarks to the Author):

In this study Daneshgar and colleagues report that a mitochondrial targeted antioxidant is able to improve disease progression in PKD.

The study has some interesting points, especially from a translational point of view. However, there is very limited novelty, there are some technical flaws (see below) and more importantly, it is unclear why the authors should ignore completely the extensive previous work published on the topic.

Reply: We appreciate the reviewer's comment and have included the citation of the relevant literature.

In the first paragraph of the discussion, we highlight five significant findings of our work: 1) mitochondrial-targeted catalase (mCAT) delivered by AAV9 significantly ameliorated the phenotypes of both PKD1 mouse models that were studied, in parallel with reducing mitochondrial ROS and tissue oxidative damage. 2) targeted proteomics studies using our PKD1 mouse models highlighted that proteins involved in FAO, TCA cycle, respiratory complexes, glucose utilization and cellular antioxidants were significantly suppressed by PKD1 mutations. 3) injection with AAV9-mCAT significantly restored, at least in part, the levels of proteins involved in FAO, endogenous antioxidants, TCA cycle and respiratory complexes and related proteins. 4) mitochondrial ROS induced ERK1/2 phosphorylation and inactivation/dephosphorylation of AMPK in human PKD mutant cell lines, whereas scavenging mitochondrial ROS attenuated ERK1/2 phosphorylation and increased AMPK phosphorylation. 5) SS31 mitochondrial protective tetrapeptide treatment mitigated the progression of ADPKD-like disease, in parallel with reducing mitochondrial ROS and oxidative damage, similar to the effects of mCAT.

Previous studies have used non-targeted antioxidants, but we focus on mitochondrial-targeted antioxidants (both mCAT and SS31). Lu et al applied transcriptomics and proteomics analysis focusing on NRF2-regulated pathways (including endogenous antioxidants). They applied unbiased TMT proteomics approaches, which usually capture more abundant proteins (data-dependent acquisition) and may not include low abundant proteins. In contrast, we applied targeted proteomics approach to detect specified targets, including many lower abundant mitochondrial and metabolic proteins. The measurements are normalized to the known amount of "spike-in" BSA. Thus, we obtained the "concentration" in pmol/ug protein, not the ratio of proteins in sample A vs. sample B. Most importantly, we demonstrate that SS31 (also known as Elamipretide) is potentially beneficial for the treatment of PKD1. Elamipretide is currently undergoing several phases II and III clinical trials. The phase I trial has proven excellent

safety profile. Thus, our study provides a scientific basis to test this new drug for the treatment of ADPKD, providing a potential addition/alternative to the expensive Tolvaptan for PKD.

Specific comments:

1. Figure 1-E the authors show a reduced mitochondrial activity and complexes assembly in Pkd mutant kidneys. It is unclear why the authors completely ignore previous studies showing similar analysis (Cassina et al, 2020). How do the results compare with the null kidneys shown in the previous study?

Reply: We have cited this important recently published paper by Cassina et al. They reported decrease in COX (Comp IV) and SDH (Comp II) by in situ enzymatic activity in *Ksp-Cre;Pkd1^{fllox/-}* mice (cystic mice), reduced mitochondrial mass and increased mitochondrial fragmentation. Mdivi-1, which interfere with Drp1, rescue mitochondrial fragmentation and ameliorates PKD1 disease progression. This study emphasizes the critical role of mitochondrial dysfunction in PKD1 and support the concept of protecting mitochondria, which is the focus of our study.

Lines 390-392 “Indeed, the concept of protecting mitochondria in PKD1 is in parallel with a recent study reporting that increased fragmentation and impaired mitochondrial function play a role in PKD1 progression³⁰.”

Despite the beneficial effect of Mdivi-1 on mitochondrial ultrastructure and PKD1 severity, it did not show any significant improvement on mitochondrial OCR in kidney samples. This is consistent with what we observed using mitochondrial-targeted catalase. We provide explanation in the discussion, as following:

Lines 405-414 “However, partial preservation of some respiratory complex subunits by mCAT apparently did not improve the overall respiratory complex activities, as measured by extracellular flux analysis (SFig. 5a-b). This may be due to the fact that some key mitochondrial encoded subunits remain present at sub-optimal levels. It is important to note that the mitochondrial electron transport activity measured here is likely far in excess of that needed to maintain ATP production. In addition, the activity measured in frozen tissue (Fig.1) relies on exogenous substrates which could be limited by the PKD1 mutation. This may explain the observation that mCAT improved the progression of PKD and the levels of many mitochondrial metabolic enzymes but does not enhance electron transport activity.”

2. In figure 2 the authors show an amelioration of the phenotype when using the AAV-mCAT vectors in the RC/RC mouse model: authors should show the data with individual points for each animal analyzed.

Is figure 2a showing a representative example or an extreme case?

Reply: Previous Figure 2a (now Fig 3a) is a representative example. The images can be correlated with the kidney weight data (fig.3b). Individual data points for all the graphs in previous Figure 2 (now as Figure 3) have been added as Supplementary Figure 2a-c.

Figure 2 d lacks indication of the Yaxis. Is BUN shown as mg/dl? Usually it has to be above 50 to be considered as a good indicator of loss of renal function. So, it is unclear what it means a value that goes from 17 to 11.

Reply: Y-axis label has been added to Fig. 2d (now as Fig. 3d). BUN has been shown as mg/dL. We agree that there is no clinical significance of the change of BUN from 17 to 11 in RC/RC, as mentioned in the manuscript. The change in BUN to > 100 in RC/null represents kidney failure (uremia) as shown in Fig. 4c.

Figure 3a is it representative or extreme case? Again, the survival curves in 3d suggest that numerous animals were analyzed, so it would be appropriate to show the individual values in figures 2b and c.

Reply: Figure 3a (now as Fig. 4a) is representative of several conducted experiments. Individual values are now presented in supplementary Fig. 3a-b. Additional images are also presented in supplementary figures.

Figure 5: TCA cycle was previously involved in PKD (Podrini et al, Comms Biol, 2018). Beta oxidation reduction was shown by two groups (Menezes et al, eBioMedicine, 2016;; Hajarnis et al, Nat Comm, 2017). And similarly mitochondrial genes.

Reply: We agree that impairment of TCA cycles, fatty acid beta oxidation and several mitochondrial genes have been described in various PKD models. In the current study, we applied targeted proteomics to demonstrate the changes of critical metabolic proteins in two PKD1 mouse models. Our data showed a gene-dosage effect of PKD1 gene on metabolic proteome. This further reinforce previously reported findings, using transcriptomics, lipidomics and metabolomics. We also show that mitochondrial-targeted antioxidant (catalase) partially restored many of these abnormalities, providing a link between protecting mitochondria and restoration of FAO and TCA in PKD1. The effect of mitochondrial targeted antioxidant has been previously studied using MitoQ in PKD1 cell line (WT9-7). In this study we further show the beneficial effect of SS31 mitochondrial protective peptide. SS31 (Elamipretide) is currently undergoing multiple clinical trials. Thus, our findings will provide a scientific basis for potential clinical trial of this drug in the future.

Indeed, involvement of mitochondria in PKD (including ROS: Lu et al, Science Translational Medicine, 2020; Irazabal and Torres, Int Journal of Molecular Sciences, 2020) has been extensively shown and the authors do not cite any of the studies. Here are some:

DOI:

10.3390/ijms21061994

10.1038/s42003-018-0200-x

10.1096/fj.201901739RR

10.1091/mbc.E16-08-0597

10.1038/s41598-018-20856-6

10.1126/scitranslmed.aba3613

Reply: The authors thank the reviewer for highlighting this point. These studies have been cited in introduction and discussion (See lines 48-50, 417-419, 433-437,449-452).

Reviewer #3 (Remarks to the Author):

The manuscript by Daneshgar et al. documents mitochondria dysfunction in ADPKD and mitochondria protective agents as potential therapeutics for ADPKD. The authors discovered a decreased mitochondrial complexes activity and downregulation of mitochondrial and metabolic proteins involved in the TCA cycle, fatty acid oxidation (FAO), respiratory complexes. Functional analysis indicated that mitochondrial-targeted catalase (mCAT) overexpression or mitochondrial protective peptide (SS31) treatment slows cystogenesis in ADPKD mouse models. Overall, this is a potentially interesting study. However, the following issue need to be solved before the publication of this manuscript can be considered.

The mitochondria defects in ADPKD are well documented in previous publications. Similar therapy (MitoQ) targeting mitochondria in human ADPKD cell model was described as in Ishimoto et al., Mol. Cell Bio., 2017 [PMID: 28993480]. The alteration of mitochondrial proteomics was also recently reported by Lu et al., Sci Transl Med., 2020 [PMID: 32727915]. Therefore, the major new finding in this study is the in vivo application of mitochondrion-targeted therapy for ADPKD. The authors show that intraperitoneal injection of AAV9-mCAT can alleviate the progression of ADPKD. However, gene delivery to kidney cells from the blood is inefficient because of the natural filtering functions of the glomerulus. The authors showed that the catalase expression was increased in kidney tubular epithelial cells with low dose injection of AAV9-mCAT (Fig. 1G). As shown in their proteomics analysis, catalase expression decreased in PKD1 mutant mice comparing to WT mice (Fig. 5A). Thus, the upregulation of

catalase may be due to disease remission rather than delivery of AAV9-mCAT to kidney cystic epithelial cells. Adding a tag to AAV9-mCAT construct and examining with anti-tag antibody would be a straightforward way to rule out the above possibility and to identify the primary target organs of AAV9-mCAT.

Reply: We agree that gene delivery to kidney cells from the blood is relatively inefficient because of the natural filtering functions of the glomerulus and the first pass effect of the liver. Our “low dose” injection of AAV9-mCAT (2×10^9 viral particles / gram body weight) is considered higher than the dosage used for local viral injection (range from 10^6 to 10^8). We have used: 1) immunofluorescence to confirm successful delivery of mCAT, which is further supported by 2) ~60% increase in catalase (by mass spectrometry measurement/proteomics). To avoid confusion, we have clarified that we used mitochondrial targeting, ornithine transcarbamylase leader sequence in the result and discussion (lines 113-115).

“we generated adeno-associated virus serotype 9 overexpressing catalase targeted to mitochondrial using ornithine transcarbamylase leader sequence (AAV9-mCAT)”

“Mitochondrial ROS has been implicated in aging and multiple diseases [see review]. To investigate the role of oxidative stress in different subcellular compartments, Schriener et al generated mice overexpressing catalase targeted to mitochondria (mCAT) or the natural sites within peroxisomes (pCAT). The beneficial effect of mCAT exceeded the effect of pCAT overexpression in term of murine lifespan extension [Science 2005], cardiac hypertrophy and failure [Cir Res 2011]. These findings emphasize the critical roles of targeting mitochondrial oxidative stress. Since then, mCAT have been shown to ameliorate various age-related diseases and cancers [reviewed in Dai et al, Progress in Molecular Biology and Translational Science Volume 146, 2017, Pages 203-241].”

Although the reviewer’s suggestion of using tag to the mCAT is excellent, however, since mCAT is targeted to mitochondria, in contrast to the natural site of catalase predominantly in cytosolic and peroxisomes, we have performed crude fractionation of RC/RC kidneys with/without AAV9-mCAT treatment and the result has been added, as following:

“In addition to colocalization of catalase and VDAC (Fig.2b), we fractionated fresh kidney tissues into mitochondrial- enriched and cytoplasmic fractions, to further confirm the effective delivery of catalase targeted to the mitochondria. Immunoblotting of catalase showed increased catalase within the mitochondrial-enriched fractions in the AAV9-mCAT treated kidneys, with the ratio of catalase in mitochondrial/cytoplasmic fractions increased by greater than 4-fold compared with that in mice treated with AAV9 vector only(SFig. 1f).”

The overexpression of mt-targeted catalase (mCAT) in the mitochondrial enriched fraction (>4-fold) is unlikely an endogenous process due to disease remission (after mCAT injection).

Furthermore, the targeted proteomics demonstrate that in the AAV9-mCAT treated kidneys, catalase is the most increased antioxidants (+64%, See mCAT effect column in Supp Table 1). Indeed, many components of glutathione peroxidase / reductase pathway (which is a natural antioxidant detoxifying hydrogen peroxide within mitochondria) were not significantly changed (Stab 1)

Co-staining with segment-specific markers would also be helpful for determining the targeting cells of mCAT.

Reply: To determine the target cells of mCAT treatment in kidneys, we performed co-staining of proximal/distal tubule specific markers (LTL/DBA, respectively) with catalase and mitochondrial marker, VDAC. As shown in supplementary fig. 1a-e, mitochondrial targeted catalase is present in both proximal and distal tubules of RC/RC and RC/null mice without obvious difference between them. Please see replies to comment 7c by reviewer #1 for further detail.

Furthermore, the authors performed immunostaining in 6-month-old RC/RC mice, in which cysts should be easily observed. However, these were not shown in their immunostaining data (Fig. 1G and 7A).

We have done additional experiments to elucidate the relationship between mCAT delivery and oxidative damage using immunofluorescence and IHC (Fig 2b, Fig 8 g-k, Sfig 1a-e). Overall, mCAT staining is evident in proximal and distal tubules, as well as some cystic lining. The nitrotyrosine staining in RC/RC kidneys is more prominent in cystic epithelium than in proximal or distal tubules. AAV9-mCAT treatment reduced the nitrotyrosine staining in proximal ($p < 0.001$), distal ($p = 0.12$) and cystic tubules ($p = 0.08$). Please see replies to comments 5d and 7c from reviewer #1 for further detail.

REVIEWERS' COMMENTS:

Reviewer #1 (Remarks to the Author):

In the revised manuscript by Daneshgar and co-workers, entitled Metabolic Derangement in PKD1 Mouse Models Is Ameliorated by Mitochondrial-Targeted Antioxidants, most of the previous concerns for this reviewer have been satisfactorily addressed by the authors with the following exceptions noted below:

1) The age of the WT, RC/null and RC/RC mice whose kidneys were used for mitochondrial respiratory complex activity in Figure 1 should be included in the Figure legend, appropriate Methods section or the Results section.

2) In Figure 6b (new data with Ki67-staining of PKD1 cells treated with MPQ or Mtmp), the figure legend notes "intensity quantification" and the y-axis of the quantification graph is labeled "fluorescence" and there is no indication that the Ki67 fluorescence has been normalized to the number of cells (i.e., Dapi-stained nuclei) present, which would be the proper method for analyzing the data. In fact, it appears from the photomicrographs that the increase (with MPQ) and decrease (with Mtemp) in Ki67 fluorescence is a reflection of increased and decreased number of cells relative to the PKD1 + saline control.

REVIEWERS' COMMENTS:

Reviewer #1 (Remarks to the Author):

In the revised manuscript by Daneshgar and co-workers, entitled Metabolic Derangement in PKD1 Mouse Models Is Ameliorated by Mitochondrial-Targeted Antioxidants, most of the previous concerns for this reviewer have been satisfactorily addressed by the authors with the following exceptions noted below:

1) The age of the WT, RC/null and RC/RC mice whose kidneys were used for mitochondrial respiratory complex activity in Figure 1 should be included in the Figure legend, appropriate Methods section or the Results section.

Reply: The age of WT, RC/null and RC/RC mice is now added to Methods section, Measurement of mitochondrial respiratory complex activity.

2) In Figure 6b (new data with Ki67-staining of PKD1 cells treated with MPQ or Mtmp), the figure legend notes "intensity quantification" and the y-axis of the quantification graph is labeled "fluorescence" and there is no indication that the Ki67 fluorescence has been normalized to the number of cells (i.e., Dapi-stained nuclei) present, which would be the proper method for analyzing the data. In fact, it appears from the photomicrographs that the increase (with MPQ) and decrease (with Mtemp) in Ki67 fluorescence is a reflection of increased and decreased number of cells relative to the PKD1 + saline control.

Reply: Positive staining for Ki67 was normalized to number of nuclei in Figure 6b. For clarification y-axis of the graph and the figure legend have been relabeled as Ki67positive/number of nuclei (%).